# Covariances for Free: Exploiting Mean Distributions for Federated Learning with Pre-Trained Models

## Abstract

Using pre-trained models has been found to reduce the effect of data heterogeneity and speed up federated learning algorithms. Recent works have investigated the use of first-order statistics and second-order statistics to aggregate local client data distributions at the server and achieve very high performance without any training. In this work we propose a training-free method based on an unbiased estimator of class covariance matrices. Our method, which only uses first-order statistics in the form of class means communicated by clients to the server, incurs only a fraction of the communication costs required by methods based on communicating second-order statistics. We show how these estimated class covariances can be used to initialize a linear classifier, thus exploiting the covariances without actually sharing them. When compared to state-of-the-art methods which also share only class means, our approach improves performance in the range of 4-26% with exactly the same communication cost. Moreover, our method achieves performance competitive or superior to sharing second-order statistics with dramatically less communication overhead. Finally, using our method to initialize classifiers and then performing federated fine-tuning yields better and faster convergence.

## 1. Introduction

Federated learning (FL) is a widely used paradigm for distributed learning from multiple clients or participants. In FL, each client trains their local model on their private data and then send model updates to a common global server that aggregates this information into a global model. The objective is to learn a global model that performs similarly to a model jointly trained on all the client data. A major concern in existing federated optimization algorithms (McMahan et al., 2017) is the poor performance when the client data is not identically and independently distributed (iid) or when classes are imbalanced between clients (Zhao et al., 2018; Li et al., 2019; Acar et al., 2021; Karimireddy et al., 2020a). Luo et al. (2021) showed that client drift in FL is mainly due to drift in client classifiers which optimize to local data distributions, resulting in forgetting knowledge from clients of previous rounds (Legate et al., 2023b; Caldarola et al., 2022). Another challenge in FL is the partial participation of clients in successive rounds (Li et al., 2019), which becomes particularly acute with large numbers of clients (Ruan et al., 2021; Kairouz et al., 2021). To address these challenges, recent works focused on algorithms to better tackle data heterogeneity between clients (Luo et al., 2021; Tan et al., 2022b; Legate et al., 2023a; Fanì et al., 2024).

Motivated by results from transfer learning (He et al., 2019), several recent works on FL have studied the impact of using pre-trained models and observe that it can significantly reduce the impact of data heterogeneity (Legate et al., 2023a; Nguyen et al., 2023; Tan et al., 2022b; Chen et al., 2022; Qu et al., 2022; Shysheya et al., 2022; Luo et al., 2021; Tan et al., 2022a). An important finding in several of these works is that sending local class means to the server instead of raw features is more efficient in terms of communication costs, eliminates privacy concerns, and is robust to gradient-based attacks (Chen et al., 2022; Zhu et al., 2019). Tan et al. (2022b) used pre-trained models to compute and then share class means as the representative of each class, and Legate et al. (2023a) showed that aggregating local means into global means and setting them as classifier weights (FedNCM) achieves very good performance without any training. FedNCM incurs very little communication cost and enables stable initialization. Recently, the authors of Fed3R (Fanì et al., 2024) explored the impact of sharing second-order feature statistics from clients to server to solve the ridge regression problem (Boyd & Vandenberghe, 2004) in federated learning and improves over FedNCM.

Fed3R communicates second-order statistics computed from local features for classifier initialization, and Luo et al. (2021) previously proposed using class means and covariances from all clients for classifier calibration after feder-

[1]Anonymous Institution, Anonymous City, Anonymous Region, Anonymous Country. Correspondence to: Anonymous Author <anon.email@domain.com>.

Preliminary work. Under review by the International Conference on Machine Learning (ICML). Do not distribute.

ated optimization. Although it is evident that exploiting second-order feature statistics results in better and more stable classifiers, it poses new problems. Notably, transferring second-order statistics for high-dimensional features from clients to the server significantly increases the communication overhead and also exposes clients to privacy risks (Luo et al., 2021; Fanì et al., 2024). In order to reap the benefits of second-order client statistics, while at the same time mitigating these risks, we propose Federated learning with COvariances for Free (FedCOF) which only communicates class means from clients to the server. We show that, from just these class means and exploiting the mathematical relationship between the covariance of class means and the class covariance matrices, we can compute an unbiased estimator of global class covariances on the server. Finally, we set the classifier weights in terms of aggregated class means and our estimated class covariances.

In this paper, we exploit pre-trained feature extractors and propose a training-free method (FedCOF) that uses the same communication budget as FedNCM while delivering performance comparable to or even superior to Fed3R. FedCOF is based on a provably unbiased estimator of class covariances that requires only class means communicated from clients to the server. We also show how to use the unbiased estimator for a better classifier initialization than Fed3R and FedNCM. We validate our proposed method across several FL benchmarks, including the real-world non-iid iNaturalist-Users-120K, and our results (see Figure 1) demonstrate that – with only a fraction of the communication costs incurred by methods communicating second-order statistics – FedCOF can achieve state-of-the-art results. Furthermore, we compare with training-based methods and show that FedCOF can be used as an initialization for federated optimization methods in order to achieve faster and better convergence.

## 2. Related Work

**Federated learning.** FL focuses on neural network training in distributed environments (Zhang et al., 2021; Wen et al., 2023). Initial works like FedAvg (McMahan et al., 2017) proposed training by averaging of distributed models. Later works focus more on non-iid settings, where data among the clients is more heterogeneous (Li et al., 2019; Kairouz et al., 2021; Wang et al., 2021; Li et al., 2021). FedNova (Wang et al., 2020) normalizes local updates before averaging to address objective inconsistency. Scaffold (Karimireddy et al., 2020b) employs control variates to correct drift in local updates. FedProx (Li et al., 2020) introduces a proximal term in local objectives to stabilize the learning process. Reddi et al. (2020) proposed use of adaptive optimization methods, such as Adagrad, Adam and Yogi, at the server side. While CCVR (Luo et al., 2021) proposed a classifier calibration by aggregating class means and covariances from clients, Li et al. (2023); Dong et al. (2022); Oh et al. (2021); Kim et al.

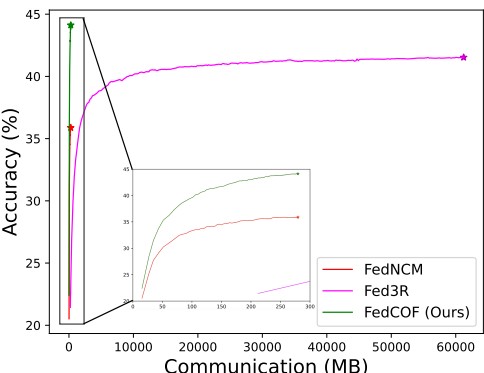

Figure 1. Performance vs. communication cost using pre-trained MobileNetv2 on iNaturalist-Users-120K. Our method (FedCOF) achieves better accuracy than Fed3R while having the same communication cost as FedNCM.

(2024) proposed using a fixed classifier motivated by the neural collapse phenomenon After federated training with fixed classifiers, FedBABU (Oh et al., 2021) proposed to fine-tune the classifiers and SphereFed (Dong et al., 2022) proposed a closed-form classifier calibration.

**FL with pre-trained models.** While conventional FL methods start training from scratch without any pre-training, we focus on the FL setting using pre-trained models. FedFN (Kim et al., 2023) recently highlighted that using pre-trained weights can sometimes negatively impact performance. However, there has been increasing interest in incorporating pre-trained, foundation models into federated learning. Multiple works propose using pre-trained weights which reduces the impact of client data heterogeneity and achieves faster model convergence (Nguyen et al., 2023; Tan et al., 2022b; Chen et al., 2022; Qu et al., 2022; Shysheya et al., 2022). Very recently, it has been shown that *training-free methods* using pre-trained networks, achieves strong performance without any training by exploiting feature class means (Legate et al., 2023a) or second-order feature statistics (Fanì et al., 2024). In this work, we propose a training-free method with pre-trained models that estimates class covariances from only client means for initializing the global classifier.

## 3. Preliminaries

### 3.1. Problem Formulation

In the FL setting, we assume $K$ clients have local datasets $D_k = (X_k, Y_k)$, where $k \in \{1, ..., K\}$. We denote the total number of images from all clients as $N$ where $N = \sum_{k=1}^{K} M_k$ and $M_k$ refers to the number of images in client $k$. We represent the model as $h_W(f_\theta(x))$ which can be decomposed into two parts: the feature extractor $f$ parameterized by $\theta$ which gives a $d$-dimensional embedding from a given image and the final classifier layer $h : \mathbb{R}^d \to \mathbb{R}^C$ parameterized by $W$ where $C$ refers to the total number of

classes. The objective of federated optimization is to learn a global model that minimizes the sum of the losses across all the clients as follows (Konečný et al., 2016) :

$$\arg\min_{\theta,W} \sum_{k=1}^{K} \frac{M_k}{N} \mathcal{L}(h_W(f_\theta(X_k)), Y_k) \quad (1)$$

where $\mathcal{L}$ is classification loss function (e.g., cross-entropy). With the growing quality of pre-trained models, recent works has focused on scenarios where all clients start with a pre-trained network (Chen et al., 2022; Legate et al., 2023a; Nguyen et al., 2023; Tan et al., 2022b; Fanì et al., 2024).

### 3.2. Training-free Federated Learning Methods

**Federated NCM.** Legate et al. (2023a) propose a Nearest Class Mean (NCM) classifier where the global linear classifier weights for class $c$ denoted by $W_c$ can be initialized as $\hat{\mu}_c/\|\hat{\mu}_c\|$ where $\hat{\mu}_c$ refers to global class means which are aggregated from the local class means $\hat{\mu}_{k,c}$ as follows:

$$\hat{\mu}_c = \frac{1}{N_c} \sum_{k=1}^{K} n_{k,c} \hat{\mu}_{k,c}; \quad \hat{\mu}_{k,c} = \frac{1}{n_{k,c}} \sum_{x \in X_{k,c}} f(x) \quad (2)$$

where $X_{k,c}$ is subset of $X_k$ having images of class $c$, $n_{k,c}$ refers to number of images in $X_{k,c}$ and $N_c = \sum_{k=1}^{K} n_{k,c}$ is the number of images of class $c$ across all clients.

**Federated Ridge Regression.** While FedNCM exploits only class means, Fed3R (Fanì et al., 2024) recently proposed to use ridge regression which needs second-order feature statistics from all clients to initialize the global classifier, leading to improved performance compared to Fed-NCM. The ridge regression problem aims to find the optimal weights that minimize the following objective:

$$W^* = \arg\min_{W \in \mathbb{R}^{d \times C}} \|Y - F^\top W\|^2 + \lambda \|W\|^2, \quad (3)$$

where $F \in \mathbb{R}^{d \times N}$ is the feature matrix extracted from a pre-trained model and $Y \in \mathbb{R}^{N \times C}$ contains one-hot encoding labels for the $N$ features with $C$ classes. The closed-form solution is given by:

$$W^* = (G + \lambda I_d)^{-1} B, \quad (4)$$

with $G = FF^\top \in \mathbb{R}^{d \times d}$ and $B = FY \in \mathbb{R}^{d \times C}$, $\lambda \in \mathbb{R}$ is an hyper-parameter and $I_d$ is the $d \times d$ identity matrix.

In Fed3R, each client $k$ computes two local matrices $G_k = F_k F_k^\top \in \mathbb{R}^{d \times d}$ and $B_k = F_k Y_k \in \mathbb{R}^{d \times C}$, where $F_k$ and $Y_k$ are the feature matrix and the labels of client $k$, and then sends them to the global server. The server aggregates these matrices as

$$G = \sum_{k=1}^{K} G_k, \quad B = \sum_{k=1}^{K} B_k \quad (5)$$

and then compute $W^*$ (Equation (4)), which is normalized and then used to initialize the global linear classifier.

*Table 1.* FedNCM (Legate et al., 2023a) shares only class means $\hat{\mu}_{k,c}$ and has minimal communication. Fed3R (Fanì et al., 2024) requires sum of class features $B_k$ and feature matrix $G_k$ from all clients, thereby increasing the communication cost by $d^2K$. We propose FedCOF, which shares only class means and estimates a global class covariance $\hat{\Sigma}_c$ to initialize the classifier weights. Here, we ignore the cost of $n_{k,c}$ which is negligible.

| Method | Client Shares | Server Uses | Comm. Cost |
|--------|--------------|-------------|------------|
| FedNCM | $\{\hat{\mu}_{k,c}, n_{k,c}\}_{c=1}^{C}$ | $\{\{\hat{\mu}_{k,c}, n_{k,c}\}_{c=1}^{C}\}_{k=1}^{K}$ | $dCK$ |
| Fed3R | $G_k, B_k$ | $\{G_k, B_k\}_{k=1}^{K}$ | $(dC + d^2)K$ |
| FedCOF | $\{\hat{\mu}_{k,c}, n_{k,c}\}_{c=1}^{C}$ | $\{\{\hat{\mu}_{k,c}, n_{k,c}\}_{c=1}^{C}\}_{k=1}^{K}, \{\hat{\Sigma}_c\}_{c=1}^{C}$ | $dCK$ |

## 4. Federated Learning with COvariances for Free (FedCOF)

### 4.1. Motivation

**Communication cost.** While Fed3R is more effective than FedNCM, it requires each client to send $C$ vectors of size $d$ and a $d \times d$ matrix, significantly increasing the communication overhead by $d^2K$ compared to FedNCM which only shares the class means (see Table 1). Fed3R scales linearly with number of clients and quadratically with the feature dimension. Smaller neural network models often have a very high-dimensional feature space. For instance, ResNet-50 has $d = 2048$ with 25.6 million parameters, MobileNetV2 has $d = 1280$ with 3.4 million parameters while ViT-B/16 has more parameters (86 million) with $d = 768$. Considering cross-device FL settings (Kairouz et al., 2021), having millions of client devices, the communication cost needed for Fed3R would be enormous. In settings with low-bandwidth communication, using Fed3R is not realistic. See Appendix I for more discussion.

**Potential privacy concerns.** Sharing only class means provides a higher level of data privacy compared to sharing raw data, as prototypes represent the mean of feature representations. It is not easy to reconstruct exact images from prototypes with feature inversion attacks (Luo et al., 2021). As a result, sharing class means is common in many recent works (Tan et al., 2022b;a; Shysheya et al., 2022; Legate et al., 2023a). On the other hand, Fed3R show that sharing second-order statistics improves the performance compared to sharing class means, but this could expose the feature distribution of clients to the server since all clients employ the same frozen pre-trained model to extract features (Fanì et al., 2024). Sharing covariances makes clients more vulnerable to attacks if secure aggregation protocols are not implemented (Bonawitz et al., 2016).

While exploiting second-order statistics (using Fed3R) yields significant gains in accuracy as shown in Figure 1, it faces the above mentioned issues. We propose instead to estimate class covariances at the server using only class means and counts from clients. This will allow us to exploit

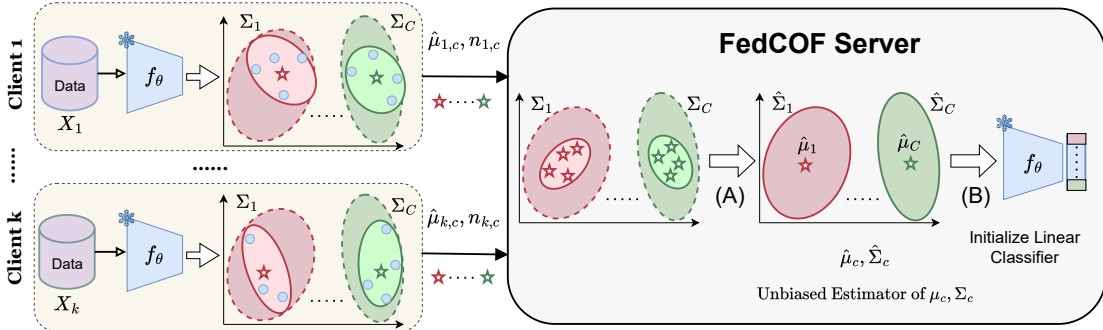

*Figure 2.* Federated Learning with COvariances for Free (FedCOF). Each client $k$ communicates only its class means $\hat{\mu}_{k,c}$ and counts $n_{k,c}$. On the server side, (A) we use a provably unbiased estimator $\hat{\Sigma}_c$ (denoted by solid lines) of population covariance $\Sigma_c$ (denoted by dashed lines) based on the received class means (see Section 4.2). (B) We initialize the linear classifier using the estimated second-order statistics and remove the between-class scatter matrix as discussed in Section 4.3.

second-order statistics without actually sharing them from clients. Following Legate et al. (2023a); Luo et al. (2021), we use class frequencies from clients since it only quantifies the client data while not revealing any information at the feature level. While this could raise minor privacy concerns, in Appendix N, we discuss methods to address those concerns.

### 4.2. Estimating Covariances Using Only Client Means

Our method leverages the statistical properties of sample means to derive an unbiased estimator of the class population covariance based only on class means (see Figure 2).

Assume that features of a class $c$ are drawn from a population with mean $\mu_c$ and covariance $\Sigma_c$. The features computed by each client are a random sample drawn from this population distribution. Using statistical properties of the sample mean we can prove the following proposition.

**Proposition 1.** *Let $\{F_{k,c}^j\}_{j=1}^{n_{k,c}}$ be a random sample from a multivariate population with mean $\mu_c$ and covariance $\Sigma_c$, where $F_{k,c}^j$ is the $j$-th feature vector of class $c$ assigned to the client $k$ and $n_{k,c}$ is the number of elements of class $c$ in the client $k$. Assuming that the per-class features $F_{k,c}^j$ in each client are iid in the initialization, then the sample mean of the features for class $c$*

$$\overline{F}_{k,c} = \frac{1}{n_{k,c}} \sum_{j=1}^{n_{k,c}} F_{k,c}^j, \qquad (6)$$

*is distributed with mean $\mathbb{E}[\overline{F}_{k,c}] = \mu_c$ and covariance $\mathrm{Var}[\overline{F}_{k,c}] = \frac{\Sigma_c}{n_{k,c}}$.*

In Appendix B we provide the proof of this well-known result about the distribution of sample means and covariances. Intuitively, since $\Sigma_c = n_{k,c}\mathrm{Var}[\overline{F}_{k,c}]$, this proposition suggests that by assigning multiple sets of $n_{k,c}$ features to a single client, we can compute the empirical covariance of the client's class means over multiple assignments, providing an estimator of population covariance $\Sigma_c$.

However, in federated learning data are assigned only once

to each client, and there are $K$ clients in the federation, each with $n_{k,c}$ features and $n_{i,c} \neq n_{j,c}$ for $i \neq j$. To estimate the population covariance $\Sigma_c$, we need an estimator that accounts for the contributions of all $K$ clients. In the following proposition, we propose such an estimator.

**Proposition 2.** *Let $K$ be the number of clients, each with $n_{k,c}$ features, and let $C$ be the total number of classes. Let $\hat{\mu}_c = \frac{1}{N_c} \sum_{j=1}^{N_c} F^j$ be the unbiased estimator of the population mean $\mu_c$ and $N_c = \sum_{k=1}^K n_{k,c}$ be the total number of features for a single class. Assuming the features for class $c$ are iid across clients at initialization, the estimator*

$$\hat{\Sigma}_c = \frac{1}{K-1} \sum_{k=1}^K n_{k,c}(\overline{F}_{k,c} - \hat{\mu}_c)(\overline{F}_{k,c} - \hat{\mu}_c)^\top \qquad (7)$$

*is an unbiased estimator of the population covariance $\Sigma_c$, for all $c \in 1, \dots, C$.*

*Proof.* To prove that $\hat{\Sigma}_c$ is an unbiased estimator of the population covariance, we show that $\mathbb{E}[\hat{\Sigma}_c] = \Sigma_c$. Under the iid assumption of client feature distribution with a frozen pre-trained model, the class features of each client can be considered as a random sample of size $n_{k,c}$, and the global class features as a sample of size $N_c$. By applying Proposition 1, we find that each client class mean has $\mathbb{E}[\overline{F}_{k,c}] = \mu_c$ and $\mathrm{Var}[\overline{F}_{k,c}] = \frac{\Sigma_c}{n_{k,c}}$, while the global class mean $\hat{\mu}_c$ has $\mathbb{E}[\hat{\mu}_c] = \mu_c$ and $\mathrm{Var}[\hat{\mu}_c] = \frac{\Sigma_c}{N_c}$. Using this fact and applying the properties of expectation to $\mathbb{E}[\hat{\Sigma}_c]$, we complete the proof. In Appendix C we provide the detailed proof. $\square$

**Covariance shrinkage.** Van Ness (1980) and Friedman (1989) proposed adding an identity matrix to the covariance matrix to stabilize the smaller eigenvalues. Shrinkage helps especially when the number of samples is fewer than the number of feature dimensions resulting in a low-rank covariance matrix. Here, the covariance estimation using a limited number of clients may poorly estimate the population covariance $\Sigma_c$. So, we perform shrinkage to better estimate

the class covariances from the client means as follows:

$$\hat{\Sigma}_c = \frac{1}{K-1} \sum_{k=1}^{K} n_{k,c}(\hat{\mu}_{k,c} - \hat{\mu}_c)(\hat{\mu}_{k,c} - \hat{\mu}_c)^\top + \gamma I_d \quad (8)$$

where $\hat{\mu}_{k,c} = \overline{F}_{k,c}$ represents a realization of client means and $\gamma > 0$ is the shrinkage factor.

**Impact of the number of clients.** The quality of estimated covariances depends on number of clients. More clients will give more means and improve the estimate compared to fewer clients. While realistic settings has thousands of clients (Hsu et al., 2020; Kairouz et al., 2021), there can be FL settings with fewer clients. In that case, we propose to sample multiple means from each client to increase number of means used for covariance estimation. This can be done by randomly sampling subsets of features in each client without replacement and computing a mean from each of these subsets. We validate this in experiments (see Figure 5).

**The iid assumption.** In FL each client has its own data, typically distributed in a statistically heterogeneous or class-imbalanced manner according to a Dirichlet distribution (Hsu et al., 2019). As a result, each client has data belonging to a different set of classes in varying quantities, resulting in non-iid data distributions across clients. However, note that the samples belonging to the same class in different clients are sampled from the same distribution. We exploit this fact in FedCOF. We later show empirically that our method can be successfully applied to non-iid FL scenarios involving thousands of heterogeneous clients on iNaturalist-Users-120K (Hsu et al., 2020). We analyze the bias of the estimator under non-iid assumptions for the same class in Appendix E and evaluate the performance of Fed-COF in *feature shift* settings (Li et al., 2021) in Appendix F.

### 4.3. Classifier Initialization with Estimated Covariances

Having derived how to compute class covariances from client means, we now discuss how to use class covariances to set the classifier weights and then replace the empirical class covariances with our estimated class covariances.

**Proposition 3.** *Let $F \in \mathbb{R}^{d \times N}$ be a feature matrix with empirical global mean $\hat{\mu}_g \in \mathbb{R}^d$, and $Y \in \mathbb{R}^{N \times C}$ be a label matrix. The optimal ridge regression solution $W^* = (G + \lambda I_d)^{-1}B$, where $B \in \mathbb{R}^{d \times C}$ and $G \in \mathbb{R}^{d \times d}$ can be written in terms of class means and covariances as follows:*

$$B = [\hat{\mu}_c N_c]_{c=1}^C, \quad (9)$$

$$G = \sum_{c=1}^{C} (N_c - 1)\hat{S}_c + \sum_{c=1}^{C} N_c(\hat{\mu}_c - \hat{\mu}_g)(\hat{\mu}_c - \hat{\mu}_g)^\top + N\hat{\mu}_g\hat{\mu}_g^\top \quad (10)$$

*where the first two terms $\sum_{c=1}^{C}(N_c - 1)\hat{S}_c$ and $\sum_{c=1}^{C} N_c(\hat{\mu}_c - \hat{\mu}_g)(\hat{\mu}_c - \hat{\mu}_g)^\top$ represents the within-class*

---

**Algorithm 1** FedCOF: FL with Covariances for Free

**Client-Side (Client $k$):**
**Input:** $C$: set of all classes, $f_\theta$: pre-trained model, $X_{k,c}$: samples of class $c$ in client $k$, $n_{k,c}$: number of samples in $X_{k,c}$
**for** $c = 1$ to $C$ **do**
$\quad \hat{\mu}_{k,c} = \frac{1}{n_{k,c}} \sum_{x \in X_{k,c}} f_\theta(x)$
**end for**
**Send** the class means $\hat{\mu}_{k,c}$ and sample counts $n_{k,c}$ to the Server

**Server-Side:**
**Input:** $\hat{\mu}_{k,c}, n_{k,c}$ sent from $K$ clients, $\lambda > 0, \gamma > 0$
**for** $c = 1 \ldots C$ **do**
$\quad \hat{\mu}_c = \frac{1}{N_c} \sum_{k=1}^{K} n_{k,c}\hat{\mu}_{k,c}; \; N_c = \sum_{k=1}^{K} n_{k,c}$ # class mean
$\quad \hat{\Sigma}_c = \frac{1}{K-1} \sum_{k=1}^{K} n_{k,c}(\hat{\mu}_{k,c} - \hat{\mu}_c)(\hat{\mu}_{k,c} - \hat{\mu}_c)^\top + \gamma I_d$, Eq.(8)
**end for**
$\hat{\mu}_g = \frac{1}{N} \sum_{c=1}^{C} N_c\hat{\mu}_c \qquad N = \sum_{c=1}^{C} N_c$ # global mean
$B = [\hat{\mu}_c N_c]_{c=1}^C$, Eq.(9)
$\hat{G} = \sum_{c=1}^{C} (N_c - 1)\hat{\Sigma}_c + N\hat{\mu}_g\hat{\mu}_g^\top$
$W^* = (\hat{G} + \lambda I_d)^{-1}B$, Eq. (11)
**Normalize** $W_c^*$: $W_c^* \leftarrow W_c^* / \|W_c^*\| \quad c = 1, \ldots, C$

---

*Table 2.* Analysis showing improved accuracy by removing between-class scatter for classifier weights initialization in centralized setting using pre-trained SqueezeNet model.

| Dataset | Using total scatter in $G$ Equation (10) | Using within-class scatter in $\hat{G}$ Equation (11) |
|---|---|---|
| CIFAR100 | 57.1 | 57.3 (+0.2) |
| ImageNet-R | 37.6 | 38.6 (+1.0) |
| CUB200 | 50.4 | 53.7 (+3.3) |
| Stanford Cars | 41.4 | 44.8 (+3.4) |

*and between class scatter respectively, while $\hat{\mu}_c$, $\hat{S}_c$ and $N_c$, denote the empirical mean, covariance and sample size for class $c$, respectively.*

*Proof.* The proof is based on the observation that $G = FF^\top$ from ridge regression is an uncentered and unnormalized empirical global covariance. By using the empirical global covariance definition and decomposing it into within-class and between-class scatter, we obtain the above formulation of $G$. In Appendix D, we provide the detailed proof.

To analyze the impact of the two scatter matrices, we consider the centralized setting in Table 2 and empirically find that using only within-class scatter matrix performs slightly better than using total scatter matrix in Equation (10). As a result, we propose to remove the between-class scatter and initialize the linear classifier at the end of the pre-trained network using the within-class covariances $\hat{\Sigma}_c$ which are estimated from client means using Equation (8), as follows:

$$W^* = (\hat{G} + \lambda I_d)^{-1}B; \quad \hat{G} = \sum_{c=1}^{C} (N_c - 1)\hat{\Sigma}_c + N\hat{\mu}_g\hat{\mu}_g^\top. \quad (11)$$

Theoretically, we observe that a similar approach is used in Linear Discriminant Analysis (Ghojogh & Crowley, 2019), which employs only within-class covariances for finding

Table 3. Evaluation of different training-free methods using 100 clients for four datasets and 9275 pre-defined clients on iNat-120K using 5 random seeds. We show the total communication cost (in MB) from all clients to server. We also show the FedCOF oracle in which full class covariances are shared from clients to server. Best results from each section in **bold**.

| | Method | SqueezeNet ($d = 512$) | | MobileNetv2 ($d = 1280$) | | ViT-B/16 ($d = 768$) | |
|---|---|---|---|---|---|---|---|
| | | Acc ($\uparrow$) | Comm. ($\downarrow$) | Acc ($\uparrow$) | Comm. ($\downarrow$) | Acc ($\uparrow$) | Comm. ($\downarrow$) |
| **CIFAR100** | FedNCM (Legate et al., 2023a) | 41.5±0.1 | **5.9** | 55.6±0.1 | **14.8** | 55.2±0.1 | **8.9** |
| | Fed3R (Fanì et al., 2024) | **56.9**±0.1 | 110.2 | 62.7±0.1 | 670.1 | **73.9**±0.1 | 244.8 |
| | FedCOF (Ours) | 56.1±0.2 | **5.9** | **63.5**±0.1 | **14.8** | 73.2±0.1 | **8.9** |
| | FedCOF Oracle (Full Covs) | 56.4±0.1 | 3015.3 | 63.9±0.1 | 18823.5 | 73.8±0.1 | 6780.0 |
| **IN-R** | FedNCM (Legate et al., 2023a) | 23.8±0.1 | **7.1** | 37.6±0.2 | **17.8** | 32.3±0.1 | **10.7** |
| | Fed3R (Fanì et al., 2024) | 37.6±0.2 | 111.9 | 46.0±0.3 | 673.1 | **51.9**±0.2 | 246.6 |
| | FedCOF (Ours) | **37.8**±0.4 | **7.1** | **47.4**±0.1 | **17.8** | 51.8±0.3 | **10.7** |
| | FedCOF Oracle (Full Covs) | 38.2±0.1 | 3645.7 | 48.0±0.3 | 22758.8 | 52.7±0.1 | 8197.4 |
| **CUB200** | FedNCM (Legate et al., 2023a) | 37.8±0.3 | **4.8** | 58.3±0.3 | **12.0** | 75.7±0.1 | **7.2** |
| | Fed3R (Fanì et al., 2024) | 50.4±0.3 | 109.6 | 58.6±0.2 | 667.3 | 77.7±0.1 | 243.1 |
| | FedCOF (Ours) | **53.7**±0.3 | **4.8** | **62.5**±0.4 | **12.0** | **79.4**±0.2 | **7.2** |
| | FedCOF Oracle (Full Covs) | 54.4±0.1 | 2472.1 | 63.1±0.5 | 15432.7 | 79.6±0.2 | 5558.6 |
| **Cars** | FedNCM (Legate et al., 2023a) | 19.8±0.2 | **5.4** | 30.0±0.1 | **13.5** | 26.2±0.4 | **8.1** |
| | Fed3R (Fanì et al., 2024) | 39.9±0.2 | 110.2 | 41.6±0.1 | 668.8 | 47.9±0.3 | 244.0 |
| | FedCOF (Ours) | **44.0**±0.3 | **5.4** | **47.3**±0.5 | **13.5** | **52.5**±0.3 | **8.1** |
| | FedCOF Oracle (Full Covs) | 44.6±0.1 | 2767.3 | 47.2±0.3 | 17275.7 | 53.1±0.1 | 6222.5 |
| **iNat-120K** | FedNCM (Legate et al., 2023a) | 21.2±0.1 | **111.8** | 36.0±0.1 | **279.5** | 53.9±0.1 | **167.7** |
| | Fed3R (Fanì et al., 2024) | 32.1±0.1 | 9837.3 | 41.5±0.1 | 61064.1 | 62.5±0.1 | 22050.2 |
| | FedCOF (Ours) | **32.5**±0.1 | **111.8** | **44.1**±0.1 | **279.5** | **63.1**±0.1 | **167.7** |
| | FedCOF Oracle (Full Covs) | 32.4±0.1 | 57k | 43.6±0.1 | 358k | 62.9±0.1 | 128k |

optimal weights. By removing between-class scatter, we propose a more effective classifier initialization than Fed3R (which uses $G$ from Equation (10) and considers both within- and between- class scatter matrices). We demonstrate this in the centralized setting (see Table 2) and in our experiments (see Table 3).

To summarize, we estimate the covariance matrix for each class using only the client means (Equation (8)) and use the estimated covariances to initialize the classifier as in Equation (11). Finally, we normalize the weights for every class to account for class imbalance in the entire dataset. We provide the summary in Algorithm 1.

**FedCOF in multiple rounds.** While the proposed estimator requires class means from all clients in a single round, this might not be realistic in settings in which clients appear in successive rounds based on availability. In the case of multiround classifier initialization (see FedCOF in Figure 3 before fine-tuning), the server uses all class means and counts received from all clients seen up to the current round and stores the accumulated means and counts for future use. As a result, FedCOF uses statistics from all clients seen up to the current round, similar to Fed3R. Thus, FedCOF converges when all clients are seen at least once. We discuss more on convergence analysis in Appendix M.

## 5. Experiments

**Datasets.** We evaluate FedCOF on multiple datasets namely CIFAR-100 (Krizhevsky, 2009), ImageNet-R (Hendrycks et al., 2021), CUB200 (Wah et al., 2011), Stanford

Cars (Krause et al., 2013) and iNaturalist (Van Horn et al., 2018). We distribute the first 4 datasets to 100 clients using a highly heterogeneous Dirichlet distribution ($\alpha = 0.1$) following standard practice (Hsu et al., 2019; Legate et al., 2023a). We also use the real-world non-iid FL benchmark of iNaturalist-Users-120K (Hsu et al., 2020) (iNat-120K) having 1203 classes across 9275 clients and 120k training images. We discuss the dataset details in Appendix H.

**Implementation Details.** We use three models: namely SqueezeNet (Iandola et al., 2016) following Legate et al. (2023a) and Nguyen et al. (2023), MobileNetV2 (Sandler et al., 2018) following Fanì et al. (2024); Hsu et al. (2020), and ViT-B/16 (Dosovitskiy et al., 2021). All models are pre-trained on ImageNet-1k (Deng et al., 2009). We use the FLSim library and implement all methods in the same framework. We use $\gamma = 1$ for all experiments with SqueezeNet and ViT-B/16, and $\gamma = 0.1$ for all experiments with MobileNetV2 due to very high dimensionality $d$ of the feature space. Following Fanì et al. (2024), we use $\lambda = 0.01$ for both Fed3R and FedCOF for numerical stability. We compare to *FedCOF Oracle* in which real class covariances are shared from clients and aggregated in server instead of using our estimated covariances (see Appendix G). For all experiments, we set the client participation in each round to 30%, and we show the training-free methods in multiple rounds in Figures 3 and 4. We provide more details in Appendix J.

### 5.1. Evaluation for different training-free methods

We compare the performance of existing training-free methods and the proposed method in Table 3 using pre-trained

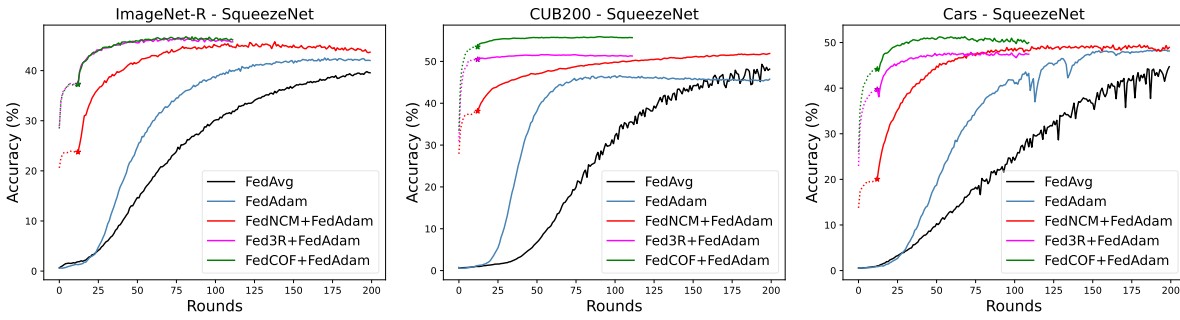

*Figure 3.* Performance comparison when initialized with different methods and then fine-tuned with FedAdam ([Reddi et al., 2020](#)) and FedAvg ([McMahan et al., 2017](#)). We also compare with FedAdam and FedAvg without any initialization (using random classifier initilization and pre-trained backbone). The training-free initialization stage for FedNCM, Fed3R and FedCOF is shown in dotted lines, star represents start of fine-tuning stage. We plot average accuracy of 3 random seeds.

Squeezenet, Mobilenetv2 and ViT-B/16 models. We observe that Fed3R ([Fanì et al., 2024](#)) using second-order statistics outperforms FedNCM ([Legate et al., 2023a](#)) significantly ranging from 0.3% to 21% across all datasets. However, Fed3R requires a higher communication cost compared to FedNCM. In real-world iNat-120K benchmark, Fed3R needs $61k$ MB compared to 280 MB for FedNCM (see Figure 1), which is 218 times higher. FedCOF performs better than Fed3R in most settings despite having the same communication cost as FedNCM. FedCOF achieves similar performance as the oracle setting using aggregated class covariances requiring very high communication, which validates the effectiveness of the proposed method.

FedCOF maintains similar accuracy with Fed3R on CIFAR100 and ImageNet-R, with an improvement of about 1% when using MobileNetv2. FedCOF outperforms Fed3R on CUB200 and Cars. On CUB200, FedCOF outperforms Fed3R by 3.3%, 3.9% and 2.2% using SqueezeNet, MobileNetv2 and ViT-B/16 respectively. FedCOF improves over Fed3R in the range of 4.1% to 5.7% on Cars. On iNat-120K, FedCOF improves over Fed3R by 0.4%, 2.6% and 0.6% using different models. When comparing FedCOF with FedNCM – both with equal communication costs and same strategy in clients – one can observe that the usage of second order statistics derived only from the class means of clients leads to large performance gains, e.g. 24% using SqueezeNet and 26% using ViT-B/16 on Cars, about 10% using all architectures on large-scale iNat-120K.

### 5.2. Comparison with training-based methods

We compare training-free methods with FL baselines like FedAvg and FedAdam with randomly initialized classifier and pre-trained backbone in Table 4. We use adaptive optimizer, FedAdam ([Reddi et al., 2020](#)) since it performs better than most other optimizers as shown in [Nguyen et al. (2023)](#). Without any training, FedCOF outperforms FedAvg in all settings and FedAdam by 7.3% on CUB200 and 2.2% on Cars, and achieves competitive performance in ImageNet-R.

*Table 4.* Comparison with training-based FL baselines (FedAvg and FedAdam) using pre-trained SqueezeNet. For training-based methods, we consider 100 rounds of training for fair comparison and report accuracy of 3 random seeds.

| Method | Training | ImageNet-R | CUB200 | Cars |
|---|---|---|---|---|
| FedAvg | ✓ | 30.0±0.6 | 30.3±6.7 | 24.9±1.6 |
| FedAdam | ✓ | 38.8±0.6 | 46.4±0.8 | 41.8±0.6 |
| FedNCM | ✗ | 23.8±0.1 | 37.8±0.3 | 19.8±0.2 |
| Fed3R | ✗ | 37.6±0.2 | 50.4±0.3 | 39.9±0.2 |
| FedCOF (Ours) | ✗ | 37.8±0.4 | 53.7±0.3 | 44.0±0.3 |
| FedNCM+FedAdam | ✓ | 44.7±0.1 | 50.2±0.2 | 48.7±0.2 |
| Fed3R+FedAdam | ✓ | 45.9±0.3 | 51.2±0.3 | 47.4±0.4 |
| FedCOF+FedAdam | ✓ | **46.0**±0.4 | **55.7**±0.4 | **49.6**±0.6 |

We show in Figure 3 how FedCOF starts from a very high accuracy compared to FedAdam and further improves on fine-tuning. We provide more experiments with pre-trained ResNet18 in Appendix K.

### 5.3. Analysis of Fine-tuning and Linear Probing

While we achieve very high accuracy without any training with FedCOF, we show in Figure 3 that further fine-tuning the model with FL optimization methods achieves better and faster convergence compared to federated optimization from scratch. We show the performance of fine-tuning after training-free classifier initialization in Figure 3. These training-free methods end after all clients appear at least once to share their local statistics to server. We fine-tune the models after FedCOF and Fed3R for 100 rounds since they achieve fast convergence, while we train for 200 rounds for FedAdam, FedAvg and fine-tuning after FedNCM which takes longer to converge. Fine-tuning after FedCOF starts with a higher accuracy and converges faster and better compared to FedNCM. Although FedCOF and Fed3R initialization converges similarly in ImageNet-R, FedCOF+FedAdam achieves a better accuracy than Fed3R+FedAdam in CUB200 and Cars. We observe in Table 4, that all training-free approaches followed by fine-tuning for 100 rounds outperform FedAdam and FedAvg with a random classifier initialization.

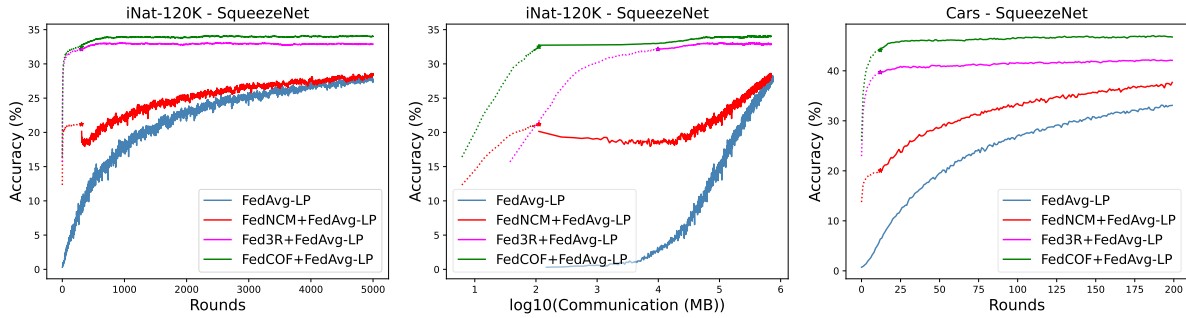

*Figure 4.* Analysis of performance when initialized with different methods and then linear-probed with FedAvg (McMahan et al., 2017). Here, FedAvg-LP (in blue) uses random classifier initialization and pre-trained backbone. The training-free initialization stage is shown in dotted lines, star represents start of linear probing stage.

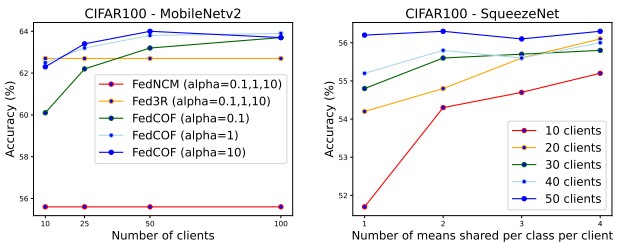

*Figure 5.* Ablation: (left) shows how performance changes with the number of clients and varying data heterogeneity; (right) shows that sharing multiple class means per client improves FedCOF performance with fewer clients.

Following Legate et al. (2023a) and Nguyen et al. (2023), we perform federated linear probing (LP) of the models using FedAvg after classifier initialization with training-free methods. In FedAvg-LP, we perform FedAvg and learn only the classifier weights of all client models. Linear probing requires much less computation compared to fine-tuning the entire model and were found to be effective with pre-trained models. We observe in Figure 4 that linear probing after FedCOF improves significantly compared to FedNCM and Fed3R using ViT-B/16 on Cars and SqueezeNet on iNat-120K. On the real-world dataset iNat-120K, FedAvg-LP with random classifier initialization achieves 27.3% after 5000 rounds while FedCOF+FedAvg-Lp achieves 34% in less than 1000 rounds. We plot accuracy vs communication in Figure 4 (middle) to demonstrate the advantage of Fed-COF over other methods. We provide more experiments in Appendix K.

### 5.4. Ablation Studies

**Impact of number of clients and data heterogeneity.** We analyze in Figure 5, the performance of FedCOF with varying number of clients and data heterogeneity. We observe that the performance of FedCOF improves with increasing number of clients and decreasing heterogeneity. This is due to the fact that more clients provides more class means and more uniform data distribution gives better representative local means. While more clients are favourable for FedCOF,

it still performs well and outperforms FedNCM significantly in the setting with 10 clients and high data heterogeneity.

**Multiple class means per client.** We analyze FL settings with fewer clients ranging from 10 to 50 in Figure 5 and show that sharing multiple class means from each client improves the accuracy. Using only 10 clients, sharing 2 class means per client improves the accuracy by 2.6%.

We discuss the impact of shrinkage hyper-parameter and present more ablation studies in Appendix L.

## 6. Conclusion

In this work, we proposed FedCOF, a novel training-free approach for federated learning with pre-trained models. By leveraging the statistical properties of client class sample means, we show that second-order statistics can be estimated using only class means from clients, thus reducing communication costs. We derive a provably unbiased estimator of population class covariances, enabling accurate estimation of a global covariance matrix. Applying shrinkage to the estimated class covariances and removing between-class scatter matrices, we show that the server can effectively use this global covariance to initialize the global classifier. Our experiments show that FedCOF outperforms FedNCM (Legate et al., 2023a) by significant margins while maintaining same communication costs. Additionally, FedCOF delivers competitive or even superior results to Fed3R (Fanì et al., 2024) across various model architectures and benchmarks while substantially reducing communication costs. Moreover, we empirically show that FedCOF can serve as a more effective starting point for improving the convergence of standard federated fine-tuning and linear probing methods.

**Limitations.** The quality of our estimator depends on number of clients, as shown in Figure 5 where using multiple class means per client helps with fewer client settings. Another limitation is the assumption that samples of the same class are iid across clients, which is, however, an assumption underlying most of federated learning. We discuss the bias in our estimator in non-iid settings in Appendix E.

Impact Statement.   In this paper we propose a highly communication-efficient method for federated learning which exploits pre-trained feature extractors. Reducing communication between clients and the central server is a critical aspect of federated learning to enhance its application in practical scenarios. The proposed method is training-free and thus does not require extensive training or incur excessive computational costs across all client devices like training-based federated learning methods. Our method drastically reduces communication while achieving similar or even better accuracy compared to existing approaches. The proposed initialization can be used with different federated fine-tuning approaches. We believe that our work will advance federated learning applications and make them more efficient.

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

## A. Scope and Summary of Notation

These appendices provide additional information, proofs, experimental results, and analyses that complement the main paper. For clarity and convenience, here we first summarize the key notations used throughout the paper:

- $N$: total number of samples.

- $K$: number of clients.

- $C$: number of classes.

- $d$: dimensionality of the feature space.

- $n_{k,c}$: number of samples from class $c$ assigned to client $k$.

- $N_c = \sum_{k=1}^{K} n_{k,c}$: total number of samples in class $c$.

- $\hat{\mu}_g, \hat{\mu}_c \in \mathbb{R}^d$: *empirical* global mean and class mean for class $c$, respectively.

- $\mu_c \in \mathbb{R}^d$: *population* mean of class $c$.

- $\hat{S}_c \in \mathbb{R}^{d \times d}$: *empirical* sample covariance for class $c$.

- $\Sigma_c \in \mathbb{R}^{d \times d}$: *population* covariance for class $c$.

- $\hat{\Sigma}_c \in \mathbb{R}^{d \times d}$: our unbiased estimator of the population covariance $\Sigma_c$ employing only client means.

- $F \in \mathbb{R}^{d \times N}$: feature matrix, where each column $F^j \in \mathbb{R}^d$ is a feature vector, for $j = 1, \ldots, N$.

- $F_{k,c}^j \in \mathbb{R}^d$: $j$-th feature vector from class $c$ assigned to client $k$.

- $\overline{F}_{k,c} \in \mathbb{R}^d$: sample mean of the feature vectors for class $c$ on client $k$, treated as a random vector. A specific realization of this random vector is denoted by $\hat{\mu}_{k,c}$.

- $\mathrm{Var}[\overline{F}_{k,c}] = \mathrm{Cov}[\overline{F}_{k,c}, \overline{F}_{k,c}]$ represents the covariance matrix of the random vector $\overline{F}_{k,c}$.

## B. Proof of Proposition 1

**Proposition 1.** *Let $\{F_{k,c}^j\}_{j=1}^{n_{k,c}}$ be a random sample from a multivariate population with mean $\mu_c$ and covariance $\Sigma_c$, where $F_{k,c}^j$ is the $j$-th feature vector of class $c$ assigned to the client $k$ and $n_{k,c}$ is the number of elements of class $c$ in the client $k$. Assuming that the per-class features $F_{k,c}^j$ in each client are iid in the initialization, then the sample mean of the features for class $c$*

$$\overline{F}_{k,c} = \frac{1}{n_{k,c}} \sum_{j=1}^{n_{k,c}} F_{k,c}^j, \tag{12}$$

*is distributed with mean $\mathbb{E}[\overline{F}_{k,c}] = \mu_c$ and covariance $\mathrm{Var}[\overline{F}_{k,c}] = \frac{\Sigma_c}{n_{k,c}}$.*

*Proof.* To prove this, we fix the class $c$ and omit the dependencies on $c$ for simplicity. Thus, we write $n_{k,c} = n_k$, $F_{k,c}^j = F_k^j$, $\overline{F}_{k,c} = \overline{F}_k$, and $\mu_c = \mu$, $\Sigma_c = \Sigma$.

Since $\{F_k^j\}_{j=1}^{n_k}$ is a random sample from a multivariate distribution with mean $\mu$ and covariance $\Sigma$, and the per-class features $F_k^j$ in each client are i.i.d at initialization, it follows that:

$$\mathbb{E}[F_k^j] = \mu \qquad \mathrm{Var}[F_k^j] = \Sigma, \quad \forall j \tag{13}$$

By computing the expectation of $\overline{F}_k$ and using the linearity of expectation, we obtain:

$$\mathbb{E}[\overline{F}_k] = \mathbb{E}[\frac{1}{n_k} \sum_{j=1}^{n_k} F_k^j] = \frac{1}{n_k} \mathbb{E}[F_k^1] + \ldots + \frac{1}{n_k} \mathbb{E}[F_k^{n_k}] = \frac{1}{n_k}(n_k \mu) = \mu,$$

where in the last equality we used Equation (13). Thus the expectation of the sample mean is $\mu$, which completes the first part of the proof.

Next, we show that the variance of the sample mean is $\frac{\Sigma}{n_k}$. By computing the variance of $\overline{F}_k$ and using the fact that the variance scales by the square of the constant, we obtain:

$$\text{Var}[\overline{F}_k] = \text{Var}[\frac{1}{n_k} \sum_{j=1}^{n_k} F_k^j] = \frac{1}{n_k^2} \left(\text{Var}[F_k^1] + \ldots + \text{Var}[F_k^{n_k}]\right) + \frac{1}{n_k^2} \sum_{i=1}^{n_k} \sum_{\substack{j=1 \\ j \neq i}}^{n_k} \text{Cov}[F_k^i, F_k^j].$$

By the independence assumption of $\{F_k^j\}_{j=1}^{n_k}$, the cross terms $\text{Cov}[F_k^i, F_k^j] = 0$ for $i \neq j$. Applying Equation (13), we have:

$$\text{Var}[\overline{F}_k] = \frac{1}{n_k^2} \left(\text{Var}[F_k^1] + \ldots + \text{Var}[F_k^{n_k}]\right) = \frac{1}{n_k^2} \left(n_k \Sigma\right) = \frac{\Sigma}{n_k}$$

$\square$

## C. Proof of Proposition 2

**Proposition 2.** *Let $K$ be the number of clients, each with $n_{k,c}$ features, and let $C$ be the total number of classes. Let $\hat{\mu}_c = \frac{1}{N_c} \sum_{j=1}^{N_c} F^j$ be the unbiased estimator of the population mean $\mu_c$ and $N_c = \sum_{k=1}^{K} n_{k,c}$ be the total number of features for a single class. Assuming the features for class $c$ are iid across clients at initialization, the estimator*

$$\hat{\Sigma}_c = \frac{1}{K-1} \sum_{k=1}^{K} n_{k,c} (\overline{F}_{k,c} - \hat{\mu}_c)(\overline{F}_{k,c} - \hat{\mu}_c)^\top \tag{14}$$

*is an unbiased estimator of the population covariance $\Sigma_c$, for all $c \in 1, \ldots, C$.*

*Proof.* To prove this, we fix the class $c$ and omit the dependencies on $c$ for clarity. So we write $n_{k,c} = n_k$, $\overline{F}_{k,c} = \overline{F}_k$, $N_c = N$, $\hat{\mu}_c = \hat{\mu}$, $\hat{\Sigma}_c = \hat{\Sigma}$, $\mu_c = \mu$, and $\Sigma_c = \Sigma$. By the definition of an unbiased estimator, we need to show that:

$$\mathbb{E}[\hat{\Sigma}] = \mathbb{E}\left[\frac{1}{K-1} \sum_{k=1}^{K} n_k (\overline{F}_k - \hat{\mu})(\overline{F}_k - \hat{\mu})^\top\right] = \Sigma.$$

By the linearity of the expectation, the definition of sample mean $\overline{F}_k = \frac{1}{n_k} \sum_{j=1}^{n_k} F_k^j$, and the definition of global class mean $\hat{\mu} = \frac{1}{N} \sum_{k=1}^{K} \sum_{j=1}^{n_k} F_k^j$, we have:

$$\mathbb{E}[\hat{\Sigma}] = \frac{1}{K-1} \left(\sum_{k=1}^{K} n_k \mathbb{E}[\overline{F}_k \overline{F}_k^\top] - \sum_{k=1}^{K} n_k \mathbb{E}[\overline{F}_k \hat{\mu}^\top] - \sum_{k=1}^{K} n_k \mathbb{E}[\hat{\mu} \overline{F}_k^\top] + \sum_{k=1}^{K} n_k \mathbb{E}[\hat{\mu} \hat{\mu}^\top]\right)$$

$$= \frac{1}{K-1} \left(\sum_{k=1}^{K} n_k \mathbb{E}[\overline{F}_k \overline{F}_k^\top] - 2\mathbb{E}[(\sum_{k=1}^{K} \sum_{j=1}^{n_k} F_k^j)\hat{\mu}^\top] + \sum_{k=1}^{K} n_k \mathbb{E}[\hat{\mu} \hat{\mu}^\top]\right)$$

$$= \frac{1}{K-1} \left(\sum_{k=1}^{K} n_k \mathbb{E}[\overline{F}_k \overline{F}_k^\top] - 2N\mathbb{E}[\hat{\mu} \hat{\mu}^\top] + \sum_{k=1}^{K} n_k \mathbb{E}[\hat{\mu} \hat{\mu}^\top]\right). \tag{15}$$

By applying the variance definition and proposition 1, we obtain:

$$\mathbb{E}[\overline{F}_k \overline{F}_k^\top] = \text{Var}[\overline{F}_k] + \mathbb{E}[\overline{F}_k]\mathbb{E}[\overline{F}_k]^\top = \frac{\Sigma}{n_k} + \mu\mu^\top. \tag{16}$$

Now, by considering the right term in Equation (15), since $\hat{\mu}$ is an unbiased estimator of the population mean, then $\mathbb{E}[\hat{\mu}] = \mu$. Moreover, since we assume that the features for a single class across clients are i.i.d at initialization, we can

re-use Proposition 1 by considering the all class features as a random sample of size $N$ from a population with mean $\mu$ and variance $\Sigma$. Consequently, the global sample mean $\hat{\mu}$ is has variance $\text{Var}[\hat{\mu}] = \frac{\Sigma}{N}$. Then

$$\mathbb{E}[\hat{\mu}\hat{\mu}^\top] = \text{Var}[\hat{\mu}] + \mathbb{E}[\hat{\mu}]\mathbb{E}[\hat{\mu}]^\top = \frac{\Sigma}{N} + \mu\mu^\top. \tag{17}$$

By using Equation (16) and Equation (17) in Equation (15), and recalling that $N = \sum_{k=1}^{K} n_k$, we obtain:

$$\mathbb{E}[\hat{\Sigma}] = \frac{1}{K-1}\left(\sum_{k=1}^{K} n_k(\frac{\Sigma}{n_k} + \mu\mu^\top) - 2N(\frac{\Sigma}{N} + \mu\mu^\top) + \sum_{k=1}^{K} n_k(\frac{\Sigma}{N} + \mu\mu^\top)\right)$$

$$= \frac{1}{K-1}(K\Sigma + \mu\mu^\top N - 2\Sigma - 2N\mu\mu^\top + (\frac{\Sigma}{N} + \mu\mu^\top)N) = \frac{1}{K-1}(K-1)\Sigma = \Sigma.$$

$\square$

## D. Proof of Proposition 3

**Proposition 3.** *Let $F \in \mathbb{R}^{d \times N}$ be a feature matrix with empirical global mean $\hat{\mu}_g \in \mathbb{R}^d$, and $Y \in \mathbb{R}^{N \times C}$ be a label matrix. The optimal ridge regression solution $W^* = (G + \lambda I_d)^{-1}B$, where $B \in \mathbb{R}^{d \times C}$ and $G \in \mathbb{R}^{d \times d}$ can be written in terms of class means and covariances as follows:*

$$B = [\hat{\mu}_c N_c]_{c=1}^{C}, \tag{18}$$

$$G = \sum_{c=1}^{C}(N_c - 1)\hat{S}_c + \sum_{c=1}^{C} N_c(\hat{\mu}_c - \hat{\mu}_g)(\hat{\mu}_c - \hat{\mu}_g)^\top + N\hat{\mu}_g\hat{\mu}_g^\top \tag{19}$$

*where the first two terms $\sum_{c=1}^{C}(N_c-1)\hat{S}_c$ and $\sum_{c=1}^{C} N_c(\hat{\mu}_c-\hat{\mu}_g)(\hat{\mu}_c-\hat{\mu}_g)^\top$ represents the within-class and between class scatter respectively, while $\hat{\mu}_c$, $\hat{S}_c$ and $N_c$, denote the empirical mean, covariance and sample size for class c, respectively.*

*Proof.* The first part, regarding Equation (18), follows directly. From the ridge regression solution, $B = FY$, which is obtained by summing the features for each class and arranging them into the columns of a matrix. This results in the product of class means and samples per class.

Now, for computing the matrix $G$, we proceed with the definition of the global sample covariance:

$$\hat{S} = \frac{1}{N-1}(F - \overline{F})(F - \overline{F})^\top = \frac{1}{N-1}\left(FF^\top - F\overline{F}^\top - \overline{F}F^\top + \overline{F}\,\overline{F}^\top\right),$$

where $\overline{F} = \left(\frac{1}{N}\sum_{j=1}^{N} F^j\right)\mathbf{1}^\top = \hat{\mu}_g\mathbf{1}^\top \in \mathbb{R}^{d \times N}$ is the matrix obtained by replicating the global mean $N$ times in each column and $\mathbf{1} \in \mathbb{R}^{N \times 1}$ is a column vector of ones. Recalling that $G = FF^\top$, we have:

$$\hat{S} = \frac{1}{N-1}(G - F\mathbf{1}\hat{\mu}_g^\top - \hat{\mu}_g\mathbf{1}^\top F^\top + \hat{\mu}_g\mathbf{1}^\top\mathbf{1}\hat{\mu}_g^\top) = \frac{1}{N-1}(G - 2F\mathbf{1}\hat{\mu}_g^\top + N\hat{\mu}_g\hat{\mu}_g^\top)$$

since $F\mathbf{1}\hat{\mu}_g^\top = \hat{\mu}_g\mathbf{1}^T F^\top$ and $\mathbf{1}^T\mathbf{1} = N$.

Now, since $F\mathbf{1} = \sum_{j=1}^{N} F^j$, we can obtain the matrix $G$ as:

$$G = (N-1)\hat{S} + 2\left(\sum_{j=1}^{N} F^j\right)\hat{\mu}_g^\top - N\hat{\mu}_g\hat{\mu}_g^\top = (N-1)\hat{S} + 2N\hat{\mu}_g\hat{\mu}_g^\top - N\hat{\mu}_g\hat{\mu}_g^\top = (N-1)\hat{S} + N\hat{\mu}_g\hat{\mu}_g^\top \tag{20}$$

It is a well known result that the global covariance can be expressed as:

$$\hat{S} = \frac{1}{N-1}\left(\sum_{c=1}^{C}(N_c - 1)\hat{\Sigma}_c + \sum_{c=1}^{C} N_c(\hat{\mu}_c - \hat{\mu}_g)(\hat{\mu}_c - \hat{\mu}_g)^T\right),$$

Replacing the global covariance $\hat{S}$ in Equation (20), we obtain the final expression for $G$ as:

$$G = \sum_{c=1}^{C}(N_c - 1)\hat{S}_c + \sum_{c=1}^{C} N_c(\hat{\mu}_c - \hat{\mu}_g)(\hat{\mu}_c - \hat{\mu}_g)^{\top} + N\hat{\mu}_g\hat{\mu}_g^{\top}$$

□

## E. Bias of the Estimator with non-iid Client Features

In Appendix C we showed that, under the assumption that the per-class features are iid across clients, the proposed estimator is an *unbiased estimator*. In this section, we theoretically quantify the bias when the i.i.d assumption is violated.

Under the i.i.d. assumption, the single class features assigned to clients can be treated as random samples from the *same* population distribution with mean $\mu_c$ and covariance $\Sigma_c$. For simplicity, focusing on a single class and dropping the class subscript $c$, the population distribution has mean $\mu$ and covariance $\Sigma$. As a result, recalling Equation (16), we can write:

$$\mathbb{E}[\overline{F}_k\overline{F}_k^{\top}] = \text{Var}[\overline{F}_k] + \mathbb{E}[\overline{F}_k]\mathbb{E}[\overline{F}_k]^{\top} = \frac{\Sigma}{n_k} + \mu\mu^{\top},$$

where $n_k$ is the number of samples assigned to client $k$, and $\overline{F}_k$ is the sample mean for client $k$

Now, if the *i.i.d assumption is violated* the local features assigned to each client can be viewed as random samples drawn from different client population distributions, each characterized by a mean $\mu_k$ and covariance $\Sigma_k$, with $\mu_i \neq \mu_j$ and $\Sigma_i \neq \Sigma_j$ for $i \neq j$, and $i, j = 1, \ldots, K$. In this case:

$$\mathbb{E}[\overline{F}_k\overline{F}_k^{\top}] = \text{Var}[\overline{F}_k] + \mathbb{E}[\overline{F}_k]\mathbb{E}[\overline{F}_k]^{\top} = \frac{\Sigma_k}{n_k} + \mu_k\mu_k^{\top}. \tag{21}$$

To compute the expectation of the estimator $\mathbb{E}[\hat{\Sigma}]$, we follow the same procedure used to prove proposition in Appendix C up to Equation (15):

$$\mathbb{E}[\hat{\Sigma}] = \frac{1}{K-1}\left(\sum_{k=1}^{K} n_k\mathbb{E}[\overline{F}_k\overline{F}_k^{\top}] - 2N\mathbb{E}[\hat{\mu}\hat{\mu}^{\top}] + \sum_{k=1}^{K} n_k\mathbb{E}[\hat{\mu}\hat{\mu}^{\top}]\right). \tag{22}$$

Assuming the global feature dataset, regardless of client assignment, is a random sample from the population with mean $\mu$ and covariance $\Sigma$, we can write:

$$\mathbb{E}[\hat{\mu}\hat{\mu}^{\top}] = \text{Var}[\hat{\mu}] + \mathbb{E}[\hat{\mu}]\mathbb{E}[\hat{\mu}]^{\top} = \frac{\Sigma}{N} + \mu\mu^{\top}. \tag{23}$$

Substituting Equation (23) and Equation (21) into Equation (22), and recalling that $N = \sum_{k=1}^{K} n_k$, we obtain:

$$
\begin{aligned}
\mathbb{E}[\hat{\Sigma}] &= \frac{1}{K-1}\left(\sum_{k=1}^{K} n_k(\frac{\Sigma_k}{n_k} + \mu_k\mu_k^{\top}) - 2N(\frac{\Sigma}{N} + \mu\mu^{\top}) + \sum_{k=1}^{K} n_k(\frac{\Sigma}{N} + \mu\mu^{\top})\right) \\
&= \frac{1}{K-1}\left(\sum_{k=1}^{K} n_k(\frac{\Sigma_k}{n_k} + \mu_k\mu_k^{\top}) - \Sigma - N\mu\mu^{\top}\right) \\
&= \frac{1}{K-1}\sum_{k=1}^{K}(\Sigma_k - \frac{\Sigma}{K}) + \frac{1}{K-1}\left(\sum_{k=1}^{K} n_k\mu_k\mu_k^{\top} - \sum_{k=1}^{K} n_k\mu\mu^{\top}\right) \\
&= \frac{1}{K-1}\sum_{k=1}^{K}(\Sigma_k - \frac{\Sigma}{K}) + \frac{1}{K-1}\sum_{k=1}^{K} n_k(\mu_k\mu_k^{\top} - \mu\mu^{\top}) \\
&= \frac{1}{K-1}\sum_{k=1}^{K}(\Sigma_k - \frac{\Sigma}{K}) + \frac{1}{K-1}\sum_{k=1}^{K} n_k(\mu_k - \mu)(\mu_k - \mu)^{\top},
\end{aligned}
$$

where in the last step we used that $\sum_{k=1}^{K} n_k\mu_k = N\mu$.

The bias of the estimator is thus given by:

$$\text{Bias}(\hat{\Sigma}) = \mathbb{E}[\hat{\Sigma}] - \Sigma = \frac{1}{K-1} \sum_{k=1}^{K} (\Sigma_k - \Sigma) + \frac{1}{K-1} \left( \sum_{k=1}^{K} n_k (\mu_k - \mu)(\mu_k - \mu)^\top \right). \tag{24}$$

Note that if each client population covariance $\Sigma_k$ is equal to the global population covariance $\Sigma$, and the mean of each client $\mu_k$ is equal to the population mean, then the bias is zero (i.e., the estimator is unbiased). However, the bias formula reveals that when the distribution of a class within a client differs from the global distribution of the same class, our estimator introduces a systematic bias. This situation can arise in the *feature-shift* setting, in which each client is characterized by a different domain. In the next section, we evaluate FedCOF under the feature-shift setting to quantify how this bias affects performance in this specific scenario.

As a final note, we mention that we always assume the global distribution of a single class can be modeled with a distribution having a single mean and covariance (see Eq. 24). This is how our classifier operates. As future work, it could be beneficial to employ different types of classifiers that allow multiple class means and class covariances.

## F. Experiments on feature shift settings

Following (Li et al., 2021), we perform experiments with MobileNetv2 in a non-iid feature shift setting on the DomainNet (Peng et al., 2019) dataset. DomainNet contains data from six different domains: Clipart, Infograph, Painting, Quickdraw, Real, and Sketch. We use the top 10 most common classes of DomainNet for our experiments following the setting proposed by (Li et al., 2021). We consider six clients where each client has i.i.d. data from one of the six domains. As a result, different clients have data from different feature distributions. We show in Table 5 how training-free methods perform in feature shift settings and the accuracy to communication trade-offs.

Table 5. Comparison of different training-free methods using MobileNetV2 on the feature shift setting on DomainNet. We show the total communication cost (in MB) from all clients to server.

| Method | Acc ($\uparrow$) | Comm. ($\downarrow$) |
|---|---|---|
| FedNCM | 65.8 | 0.3 |
| Fed3R | 81.9 | 39.6 |
| FedCOF | 74.1 | 0.3 |
| FedCOF (2 class means per client) | 76.5 | 0.6 |
| FedCOF (10 class means per client) | 78.8 | 3.1 |

Fed3R achieves better overall performance then FedCOF, likely due to its use of exact class covariance, avoiding the bias that FedCOF introduces. However, FedCOF achieves comparable results while significantly reducing communication costs. FedNCM perform worse than FedCOF at the same communication budget. When we increase the number of means sampled from each client, the performance of our approach improves. This is due to the fact that our method suffers with low number of clients (only 6 in this experiments) and sampling multiple means helps.

## G. The FedCOF Oracle (Sharing Full Covariances)

Similar to (Luo et al., 2021), we aggregate the class covariances from clients as follows:

$$\hat{\Sigma}_c = \sum_{k=1}^{K} \frac{n_{k,c} - 1}{N_c - 1} \hat{\Sigma}_{k,c} + \sum_{k=1}^{K} \frac{n_{k,c}}{N_c - 1} \hat{\mu}_{k,c} \hat{\mu}_{k,c}^T - \frac{N_c}{N_c - 1} \hat{\mu}_c \hat{\mu}_c^T. \tag{25}$$

We use the aggregated class covariance from Equation (25) and apply shrinkage to obtain $\hat{\Sigma}_c + \gamma I_d$ and use it in Equation (11) for the oracle setting of FedCOF.

## H. Dataset Details

CIFAR-100 has 100 classes provided in 50k training and 10k testing images. ImageNet-R (IN-R) is composed of 30k images covering 200 ImageNet classes. ImageNet-R (Hendrycks et al., 2021) is an out-of-distribution dataset and proposed to evaluate out-of-distribution generalization using ImageNet pre-trained weights. It contains data with multiple styles like cartoon, graffiti and origami which is not seen during pre-training. We also consider fine-grained datasets like CARS and CUB200 for our experiments. CUB200 has as well 200 classes of different bird species provided in 5994 training and 5794

testing images. Stanford Cars has 196 classes of cars with 8144 training images and 8041 test images. Finally, we also use iNaturalist-Users-120k (Hsu et al., 2019) dataset in our experiments, which is a real-world, large-scale dataset (Van Horn et al., 2018) proposed by (Hsu et al., 2019) for federated learning and contains 120k training images of natural species taken by citizen scientists around the world, belonging to 1203 classes spread across 9275 clients. In datasets like ImageNet-R and CARS, we also face class-imbalanced situations where there is a significant class-imbalance at the global level.

## I. Communication Costs

We show in Figure 6 how the communication cost of Fed3R increases drastically as the dimensionality of the feature space increases and the number of clients increases since Fed3R needs to share high dimensional second-order statistics from clients to server. On the other hand, our proposed method FedCOF has the same communication costs of FedNCM and scales linearly with the feature dimensionality and the number of clients.

When computing communication costs we consider that the pre-trained models are on the clients and do not need to be communicated. We do not include cost of backward communication of classifier parameters from server to clients, since it is the same for all methods but is necessary only if the models are fine-tuned after classifier initialization. All parameters are considered to be 32-bit floating point numbers (i.e. 4 bytes) in all our analysis and experiments.

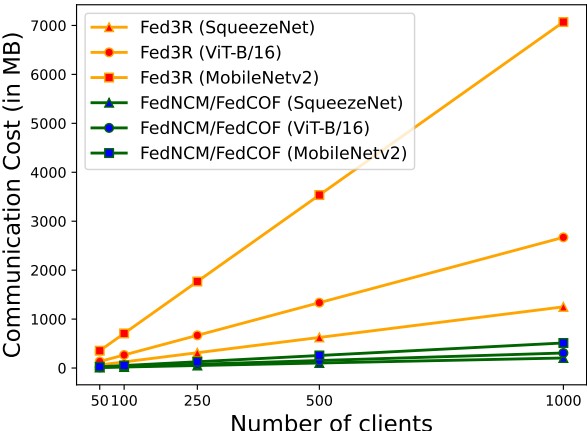

*Figure 6.* Analysis showing increasing communication cost for Fed3R with increasing number of clients assuming 100 classes per client. This is due to the high dimensionality of the features ($d = 512$ for SqueezeNet, $d = 768$ for ViT-B/16 and $d = 1280$ for MobileNetV2).

## J. Implementation Details

Here, we provide details on learning rate (lr) used for all fine-tuning experiments with FedAdam. For ImageNet-R and Stanford Cars, we use a lr of 0.0001 for both server and clients for FedNCM, Fed3R and FedCOF initializations. For CUB200, we use a server lr of 0.00001 and client lr of 0.00005 for Fed3R and FedCOF, while for FedNCM, we use a higher lr of 0.0001 for clients. For random classifier initialization with all datasets, we use a higher lr of 0.001 for clients and lr of 0.0001 for server. We use 1 local epoch, for all fine-tuning experiments on 4 datasets. After training-free classifier initialization, we fine-tune the models for 100 rounds. When starting from random classifier initialization, we train more for 200 rounds. When training with FedAvg and random classifier initialization, we use a client lr of 0.005 for all datasets other than inat-120K.

For the linear probing experiments for 4 datasets other than inat-120K, with FedAvg we train for 200 rounds with 1 local epoch and use a client lr of 0.01 and server lr of 1.0 for FedNCM. For Fed3R and FedCOF initializations, we use a client lr of 0.001 and a server lr of 1.0. For LP experiments on iNat-120K, we use 3 local epochs, 30% client participation and train for 5000 rounds. For iNat-120K, we use a client lr of 0.001 for FedAvg-LP without classifier initialization, client lr of 0.0005 for FedNCM and client lr of 0.00001 for Fed3R and FedCOF.

We use Nvidia RTX 6000 GPU for our experiments. We will make the code publicly available for reproducing our results for all experiments.

## K. Additional Experiments

**Linear probing after initialization experiments.** We show in Figure 7 that linear probing after FedCOF classifier initialization improves the accuracy significantly compared to FedNCM and is marginally better than Fed3R initialization across three datasets using SqueezeNet.

**Comparison of training-free methods with linear probing.** We also compare with our approach with the training-based federated linear probing without any initialization (where we perform FedAvg and learn only the classifier weights of models) and show in Table 6 that FedCOF is more robust and communication-efficient compared to federated linear probing across several datasets. We follow the same settings as in Table 3. For first 4 datasets, we perform federated linear probing

*Figure 7.* Analysis of the performance with federated linear probing using FedAvg (McMahan et al., 2017).

*Table 6.* Comparison of different training-free methods using SqueezeNet with training-based Fed-LP (federated linear probing with FedAvg (McMahan et al., 2017) starting with pre-trained model and random classifier initialization) across 5 random seeds. FedNCM, Fed3R and the proposed FedCOF does not involve any training. We show the total communication cost (in MB) from all clients to server. The best results from each section are highlighted in **bold**.

| Method | CIFAR100 | | ImageNet-R | | CUB200 | | CARS | | iNat-120K | |
|---|---|---|---|---|---|---|---|---|---|---|
| | Acc (↑) | Comm. (↓) | Acc (↑) | Comm. (↓) | Acc (↑) | Comm. (↓) | Acc (↑) | Comm. (↓) | Acc (↑) | Comm. (↓) |
| Fed-LP | **59.9**±0.2 | 2458 | **37.8**±0.3 | 4916 | 46.8±0.8 | 4916 | 33.1±0.1 | 4817 | 28.0±0.6 | $1.6\times10^6$ |
| FedNCM | 41.5±0.1 | **5.9** | 23.8±0.1 | **7.1** | 37.8±0.3 | **4.8** | 19.8±0.2 | **5.4** | 21.2±0.1 | **111.8** |
| Fed3R | 56.9±0.1 | 110.2 | 37.6±0.2 | 111.9 | 50.4±0.3 | 109.6 | 39.9±0.2 | 110.2 | 32.1±0.1 | 9837.3 |
| FedCOF (Ours) | 56.1±0.2 | **5.9** | **37.8**±0.4 | **7.1** | **53.7**±0.3 | **4.8** | **44.0**±0.3 | **5.4** | **32.5**±0.1 | **111.8** |

for 200 rounds with 30 clients per round using FedAvg with a client learning rate of 0.01. For iNat-120k, we train more for 5000 rounds.

**Impact of using pre-trained models.** To quantify impact of using pre-trained models we performed experiments using a randomly initialized model and show in Table 8 that federated training using a pre-trained model significantly outperforms a randomly initialized model using standard methods like FedAvg and FedAdam on CIFAR-10 and CIFAR-100.

**Experiments with ResNet18.** We perform experiments with pre-trained ResNet18 in Table 7. For FedAvg and FedAdam, we train for 200 rounds with 30 clients per round. For FedAvg, we train with a client learning rate of 0.001 and server learning rate of 1.0. For FedAdam, we train with a client learning rate of 0.001 and a server learning rate of 0.0001. We show that fine-tuning after FedCOF classifier initialization for 100 rounds outperforms competitive FL methods like FedAdam (which are trained for 200 rounds) by 2.5% on CIFAR100 and 5.1% on ImageNet-R. The improved performance with FedCOF initialization validates the effectiveness of the proposed method, as it reduces communication and computation costs by half compared to FedAdam and FedAvg and still outperforms them.

## L. Additional Ablations

**Impact of Shrinkage.** We analyze the impact of using shrinkage on the estimated class covariances in the proposed method FedCOF using pre-trained SqueezeNet in Table 9. We use a shrinkage $\gamma = 1$ for our experiments with SqueezeNet and ViT-B/16. We observe that shrinkage has marginal improvement for ImageNet-R and a bit more significant improvement in accuracy by 2.5% on CUB200. This observation can be

*Table 9.* Ablation showing the impact of using shrinkage in Fed-COF using pre-trained SqueezeNet.

| Dataset | $\gamma = 0$ | $\gamma = 0.01$ | $\gamma = 0.1$ | $\gamma = 1$ | $\gamma = 10$ |
|---|---|---|---|---|---|
| ImageNet-R | 36.53 | 36.98 | 36.96 | 37.25 | 36.07 |
| CUB200 | 51.08 | 51.07 | 51.81 | 53.57 | 53.50 |

attributed to the few-shot settings where the covariance estimation is not very good owing to lack of data and thus lesser clients having access to each of the classes. The use of shrinkage in FedCOF stabilizes and improves the covariance estimation leading to improved accuracy especially in few-shot settings.

**Sampling multiple class means.** We perform multiple class means sampling per client using ImageNet-R and show in Figure 8 (left) that using FedCOF with more class means shared from each client improves the performance. We also show in Figure 8 (middle) the total number of means used per class on an average in Figure 8 (left) to perform the covariance

*Table 7.* Comparison of different training-free methods using pre-trained ResNet18 for 100 clients with training-based federated learning baselines FedAvg (McMahan et al., 2017) and FedAdam (Reddi et al., 2020) starting from a pre-trained model. We train for 200 rounds for FedAvg and FedAdam which uses pre-trained backbone and random classifier initialization. FedNCM, Fed3R and the proposed FedCOF do not involve any training. We also show the performance of fine-tuning with FedAdam after classifier initialization. For fine-tuning experiments we only train for 100 rounds after initialization. We show the total communication cost (in MB) from all clients to server. The best results from each section are highlighted in **bold**.

| Method | CIFAR100 | | ImageNet-R | |
|---|---|---|---|---|
| | Acc ($\uparrow$) | Comm. ($\downarrow$) | Acc ($\uparrow$) | Comm. ($\downarrow$) |
| FedAvg | 67.7 | 538k | 56.0 | 541k |
| FedAdam | 74.4 | 538k | 57.1 | 541k |
| FedNCM | 53.8 | **5.9** | 37.2 | **7.1** |
| Fed3R | 63.5 | 110.2 | 45.9 | 111.9 |
| FedCOF (Ours) | 63.3 | **5.9** | 46.4 | **7.1** |
| FedNCM+FedAdam | 75.7 | 269k | 60.3 | 271k |
| Fed3R+FedAdam | 76.8 | 269k | 60.6 | 271k |
| FedCOF+FedAdam | **76.9** | 269k | **62.2** | 271k |

*Table 8.* Impact of using pre-trained SqueezeNet network with different federated learning methods on CIFAR10 and CIFAR100. We show the total communication cost (in MB) from all clients to server. We train 100 clients with 30 clients per round for 200 rounds in non-iid settings with dirichlet distribution of 0.1. When starting from random initialization (no pre-training), we train for 400 rounds.

| Method | Pre-trained | CIFAR10 | | CIFAR100 | |
|---|---|---|---|---|---|
| | | Acc ($\uparrow$) | Comm. ($\downarrow$) | Acc ($\uparrow$) | Comm. ($\downarrow$) |
| FedAvg | $\times$ | 37.3 | 74840 | 23.9 | 79248 |
| FedAdam | $\times$ | 60.5 | 74840 | 44.3 | 79248 |
| FedAvg | $\checkmark$ | 84.7 | 37420 | 56.7 | 39624 |
| FedAdam | $\checkmark$ | 85.5 | 37420 | 62.5 | 39624 |

estimation. The number of means used to estimate each class covariance is less than the total number of clients due to the class-imbalanced or dirichlet distribution used to sample data for clients. This is due to the fact that not all classes are present in all clients.

**Communicating diagonal or spherical covariances.** While communicating diagonal or spherical covariances (mean of the diagonal covariance) from clients to server and then estimating the global class covariance from them can significantly reduce the communication cost, such estimates of global class covariance is poor compared to FedCOF. We show in Figure 8 (right) that FedCOF outperforms these covariance sharing baselines when communicating spherical or diagonal covariances.

## M. Convergence Analysis

In our work, we claim that FedCOF initialization achieves faster and better convergence based on our empirical results (Figure 3) using multiple datasets. We propose how to initialize the classifiers before performing federated optimization methods like FedAvg (McMahan et al., 2017) and FedAdam (Reddi et al., 2020) which have already established the theoretical guarantees of convergence in their respective works. Unlike gradient-based FL methods, our method is training-free. Similar to Fed3R (Fanì et al., 2024) and FedNCM (Legate et al., 2023a), the proposed FedCOF does not depend on assumptions like bounded variance of stochastic gradients or smoothness of clients objectives. Similar to exisiting training-free methods like FedNCM and Fed3R, FedCOF uses statistics from all clients seen up to the current round. As a result, all these training-free methods including FedCOF converges when all clients are seen at least once.

While we do not propose any federated optimization step, we propose a training-free method that can be also used for initializing federated fine-tuning. We would also like to highlight that all existing works in Federated learning with pre-trained models (Tan et al., 2022b; Nguyen et al., 2020; Fanì et al., 2024; Legate et al., 2023a; Chen et al., 2022; Qu et al., 2022; Shysheya et al., 2022) focus only on empirical observations assuming that the theoretical guarantees of existing federated optimization methods holds true when using pre-trained models. A more exhaustive study on convergence analysis for FL with pre-trained models would be an interesting direction to explore in future works.

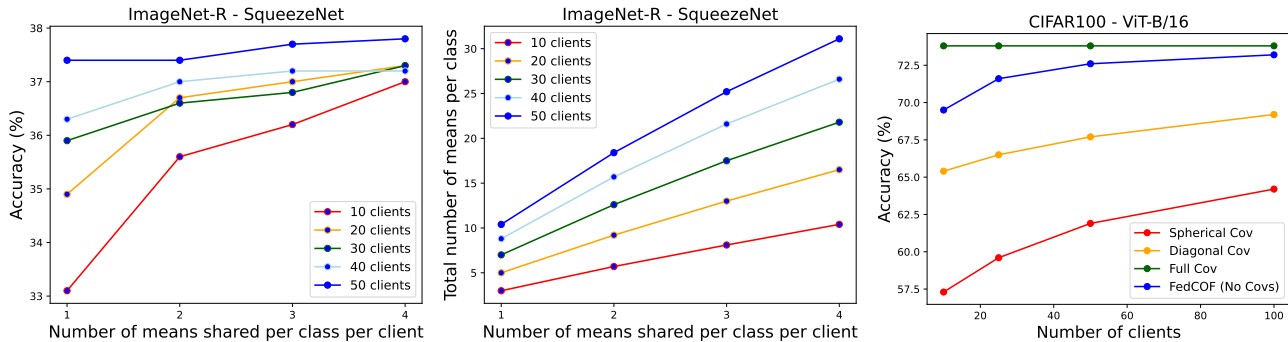

*Figure 8.* (left) Analysis of FedCOF performance with multiple class means per client on ImageNet-R. (middle) Total number of means per class on average that are used to estimate the covariance for FedCOF in Figure 8 (left). (right) Performance comparison of FedCOF with full, diagonal, and spherical covariance matrix communication.

## N. Privacy Concerns on sharing Class-wise Statistics

Our method requires transmitting class-wise statistics to compute the unbiased estimator of the population covariance (Equation (14)) and classifier initialization, similar to other methods in federated learning (Legate et al., 2023a; Luo et al., 2021). In general, transmitting the class-wise statistics may raise privacy concerns, since each client could potentially expose its class distribution. Inspired by differential privacy (Dwork et al., 2006), we propose perturbing the class-wise statistics of each client with different types and intensities of noise, before transmission to the global server. This analysis allows us to evaluate how robust FedCOF is to variations in class-wise statistics and whether noise perturbation mechanisms can effectively hide the true client class statistics. Specifically, we propose perturbing the class-wise statistics as follows:

$$\widetilde{n}_{k,c} = \max(n_{k,c} + \sigma_\epsilon^{\mathrm{noise}}, 0) \tag{26}$$

where $\sigma_\epsilon^{\mathrm{noise}}$ is noise added to the statistics, and $\epsilon$ is a parameter representing the noise intensity. The $\max$ operator clips the class statistics to zero if the added noise results in negative values, which is expected to happen in federated learning with highly heterogeneous client distributions. When clipping is applied, the client does not send the affected class statistic and class mean, and the server excludes them from the computation of the unbiased estimator.

We consider three types of noise:

- *Uniform noise*: $\sigma_\epsilon^{\mathrm{unif}} \sim \mathcal{U}(-(1-\epsilon)n_{k,c}, +(1-\epsilon)n_{k,c})$, proportional to the real class statistics.

- *Gaussian noise*: $\sigma_\epsilon^{\mathrm{gauss}} \sim \mathcal{N}(0, \frac{1}{\epsilon})$, independent of the real class statistics.

- *Laplacian noise* $\sigma_\epsilon^{\mathrm{laplace}} \sim \mathcal{L}(0, \frac{1}{\epsilon})$, which is also independent of the real class statistics.

Lower $\epsilon$ values correspond to higher levels of noise in the statistics.

In Figure 9, we show that the performance of FedCOF is robust with respect to the considered noise perturbation, varying the intensity of $\epsilon \in \{0.1, 0.3, 0.5, 0.7, 0.9\}$. These results suggest that a differential privacy mechanism can be implemented to mitigate privacy concerns arising from the exposure of client class-wise frequencies. In Figure 10, we provide a qualitative overview of how the proposed Laplacian and uniform noise perturbation affect class-wise distributions.

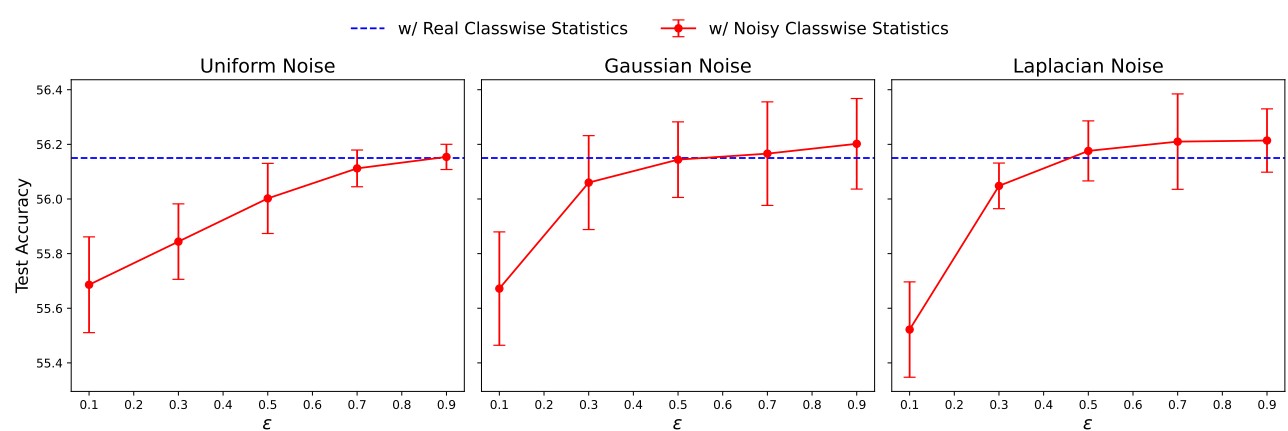

*Figure 9.* Performance of FedCOF with noisy class statistics on CIFAR-100 using SqueezeNet. The number of clients is fixed at 100 and classes are distributed using a Dirichlet distribution with $\alpha = 0.1$. Results are averaged over five random seeds, each generating different noise in client statistics, and the standard deviation is reported. FedCOF demonstrates robustness to uniform, Gaussian, and Laplace perturbations in class statistics, with performance showing a slight drop as noise, parameterized by $\epsilon$, increases. Lower $\epsilon$ corresponds to higher noise levels in the class statistics.

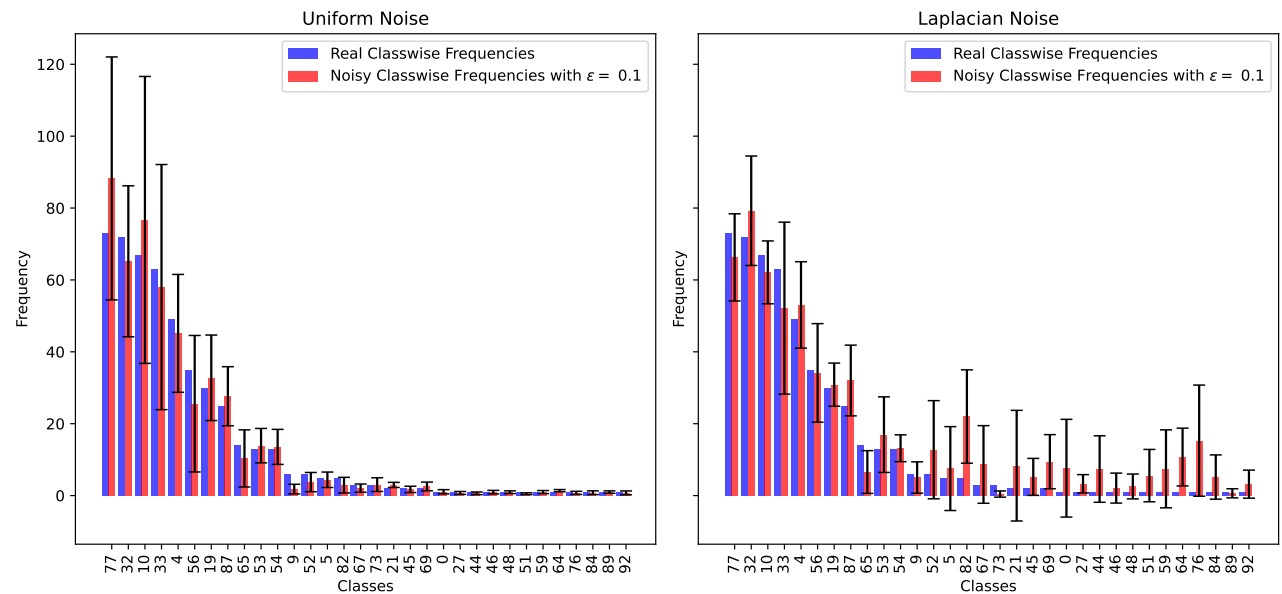

*Figure 10.* Class frequency distributions for a single client under different noise types: uniform noise (left) and Laplacian noise (right) on CIFAR-100. Both noise types are applied to the real class statistics with the highest noise intensity ($\epsilon = 0.1$). The bar heights represent the average class frequencies, and the error bars indicate the standard deviation across 5 seeds. Real class-wise frequencies and their noisy counterparts are shown for comparison.

