# OpenReview forum: "Covariances for Free: Exploiting Mean Distributions for Federated Learning with Pre-trained Models"
_ICML.cc/2025/Conference — Submitted to ICML 2025_

### Official Review · Reviewer_FJXY · 2025-03-11

**Overall Recommendation:** 3

**Summary:**

The paper presents a novel training-free FL method, starting from well-pretrained model as an initialization. The authors tackle the key limitation of the existing work, Fed3R, where the clients must upload the second order statistics, incurring additional communication cost. To address this issue, the authors propose to use  an unbiased estimation of the second order statistics by using only the mean estimator. Furthermore, the authors remove between-class scatter in the estimation of $G$, which turns out to be more effective than Fed3R classifier. Experimental results demonstrate that their proposed approach achieves better accuracy than the baselines in most settings with less communication burden.

**Claims And Evidence:**

The claims made in the paper seem to be well supported by extensive experimental results and analyses.

**Essential References Not Discussed:**

I have not noticed any significant prior works that were not discussed.

**Experimental Designs Or Analyses:**

The experimental settings and evaluation metrics are well designed and valid.

**Methods And Evaluation Criteria:**

The method is conceptually sound, and the evaluation criteria make sense.

**Other Comments Or Suggestions:**

It appears that the number of hyperparmaters ($\gamma$ and $\lambda$) is reduced to one, as the covariance shrinkage can be absorbed into $\lambda I_d$ in equation (11). In addition, it would be helpful to elaborate on why the removal of between-class scatter in the estimation of $G$ leads to performance improvement in the FL setup.

**Other Strengths And Weaknesses:**

### Strengths

- The paper is well structured and clearly written.
- The paper significantly reduces the communication cost compared to previous work by uploading only the mean estimator while also improving the performance by removing between-class scatter.
- The proposed covariance estimation using a mean estimator is mathematically well justified and conceptually solid.
- The extensive simulations including various ablations and analyses are provided, which further strengthen the validity of the paper.

### Weaknesses

- While the performance improvement is promising, there is no clear justification for removing between-class scatter in FL. It seems that the authors made this choice primarily based on its empirical gain observed in the centralized setup.
- The reliability of estimator heavily relies on the number of clients.

**Questions For Authors:**

See weaknesses and comments.

**Relation To Broader Scientific Literature:**

One of the key contributions of this paper lies in reducing communication costs compared to previous method by transmitting only the mean estimator instead of real second order statistics. Furthermore, they explore the removal of between-class scatter, which has proven to be more effective than the previous classifier.

**Theoretical Claims:**

The proposed covariance estimation using a mean estimator is grounded in mathematical rigor.

---

> ### Author Rebuttal · Authors · 2025-03-31
>
> We thank the reviewer for appreciating that the paper is well structured and clearly written, that claims are well supported by extensive experimental results and analyses, that the covariance estimator is mathematically well justified and conceptually solid, and that the experimental settings are well designed and valid. Below we reply to the specific points raised by the reviewer.
>
> >While the performance improvement is promising, there is no clear justification for removing between-class scatter in FL.
>
> The reviewer is right, we made the choice to remove between-class scatter primarily based on empirical results in the centralized setup because the Ridge Regression solution is equivalent in the centralized and federated learning setups (as discussed also in Fed3R). While we propose this based on empirical results, we provide more justification and analysis on this below.
>
> Intuitively, to represent the feature distribution of each class we do not really need to consider the relationships between different classes which represent the distribution of the overall dataset since our goal is to estimate the *class-specific classifier weights* using these covariances. Based on this, we propose to remove the between-class scatter and initialize the classifier weights using only the respective within-class scatter matrix.
>
> Let us denote the between- and within-class scatter matrices as $G_{\text{btw}}$ and $G_{\text{with}}$, respectively. We analyze the spectral properties of these scatter matrices, with a focus on their conditioning. By defining the condition number for a matrix $G$ as  $k(G) =  \lambda_{\text{max}}(G) / \lambda_{\text{min}}^{+}(G) $,  where $\lambda_\text{max}$ is the largest eigenvalue and $\lambda_\text{min}^{+}(G)$ is the smallest non-zero eigenvalue, we observe empirically for SqueezeNet in the centralized setting that:
> - $G_{\text{btw}}$ is severly ill-conditioned with condition numbers $k(G_{\text{btw}})$: $2.97\times10^7$, $2.46\times 10^7$,$2.2 \times 10^7$, $1.27\times10^7$ on CUB, CARS, ImageNet-R, CIFAR-100, respectively.
> - $G_{\text{with}}$ is much better conditioned with condition numbers $k(G_{\text{with}})$: $4.5\times10^3$, $2.48\times10^4$, $8.19 \times 10^2$, $6.3 \times 10^3$ on CUB, CARS, ImageNet-R, CIFAR-100.
>
> Including $G_{\text{btw}}$, can cause numerical instability because it is poorly conditioned. This leads the classifier to overfit to directions with very small eigenvalues, which may capture noise or dataset-specific artifacts, resulting in poor generalization on unseen samples.
>
> We also analyze in the centralized setting how the different scatter matrices affect overfitting of the model. We fix the parameter $\lambda=0.01$, which yields optimal performance for both classifiers.
>
> |Classifier|$G_{\text{with}}$|$G_{\text{btw}}$|Dataset|Train Acc|Test Acc|
> |-|-|-|-|-|-|
> |Ridge Regression|✅|✅|CUB|92.0|50.4|
> |Ours|✅|❌|CUB|91.3|**53.7**|
> ||
> |Ridge Regression|✅|✅|Cars|85.9|41.4|
> |Ours|✅|❌|Cars|86.2|**44.8**|
> ||
> |Ridge Regression|✅|✅|IN-R|52.8|37.6|
> |Ours|✅|❌|IN-R|53.4|**38.6**|
> ||
> |Ridge Regression|✅|✅|Cifar100|60.0|57.1|
> |Ours|✅|❌|Cifar100|60.4|**57.3**|
>
> We observe that, while both methods achieve similar high training accuracies, Ridge Regression consistently underperforms on the test set. This suggests that incorporating $G_{\text{btw}}$ introduces overfitting, as the classifier learns directions that do not transfer well to unseen samples.
>
> Finally, we analyze the impact of removing the $G_{\text{btw}}$ in a FL setup using SqueezeNet below:
>
> ||CIFAR100|IN-R|CUB|CARS|
> |-|-|-|-|-|
> |Using $G_{\text{btw}}$|52.8|34.8|49.7|33.7|
> |Using $G_{\text{btw}}$+$G_{\text{with}}$|56.3|36.8|51.6|42.4|
> |Using $G_{\text{with}}$ (Ours)|56.3|37.2|53.5|44.6|
>
> In the FL setup, we again clearly see the negative effect of incorporating between-class scatter statistics. We will add all the analysis and discussion in the revised version of the paper.
>
> >The reliability of estimator heavily relies on the number of clients.
>
> We acknowledge this in the Limitations section. The quality of our estimator depends on number of clients, as shown in Fig. 5 where using *multiple* class means per client helps in fewer client settings. In Appendix L (Fig. 8) we explain this in more details.
>
> >It appears that the number of hyperparmaters ($\gamma$ and $\lambda$) is reduced to one, as the covariance shrinkage can be absorbed into $\lambda I_d$  in equation (11).
>
> The reviewer is correct, the two hyperparameters have similar purposes and can be absorbed into one. While we used $\lambda$ to maintain the same formulation as the ridge regression classifier, our method does not require this hyperparameter. Varying the $\lambda$ parameter, we do not observe any change in the performance of FedCOF. Thus, the $\lambda$ hyperparameter can be removed since the shrinkage hyperparameter already serves the same purpose.

---

### Official Review · Reviewer_tc2T · 2025-03-18

**Overall Recommendation:** 3

**Summary:**

The main conceptual idea is to estimate class covariance matrices at the server using only class means communicated from clients, avoiding the need to share computationally expensive second-order statistics (e.g., covariance matrices) as in prior methods like Fed3R. FedCOF exploits the statistical relationship between the covariance of class means and population class covariances to derive an unbiased estimator, which is then used to initialize a linear classifier. Key contributions include: (1) a provably unbiased estimator of class covariances requiring only first-order statistics (class means), (2) a significant reduction in communication overhead compared to methods relying on second-order statistics, and (3) improved performance over existing training-free methods like FedNCM (4-26% accuracy gains) and competitive or superior results compared to Fed3R with lower communication costs.

**Claims And Evidence:**

- The covariance estimator is unbiased under the iid assumption. Proposition 2 and its proof in Appendix C provide a mathematical foundation, showing that the expectation of the estimator equals the population covariance. However, the evidence weakens when the iid assumption is relaxed (Appendix E), where bias is acknowledged but not quantified beyond theoretical derivation, leaving practical implications partially unsupported.

- Justification of using only within class covariance terms are supported only by numerical results (ablative study in Table 2), and this methods underperforms Fed3R in the presence of feature shift. More extensive analysis and justification with theoretical evidence is required for the proposed method.

**Essential References Not Discussed:**

key related works are discussed properly

**Experimental Designs Or Analyses:**

Training-Free Evaluation (Table 3): Compares FedCOF against FedNCM and Fed3R across five datasets and three models, using accuracy and communication cost metrics. The design is sound, with 5 random seeds ensuring statistical reliability, and results are consistent across settings. The FedCOF Oracle baseline validates the estimator’s quality.

Fine-Tuning and Linear Probing (Figures 3, 4, Tables 4, 6): Assesses FedCOF as an initialization for FedAvg/FedAdam. The use of 100-200 rounds (fine-tuning) and up to 5000 rounds (linear probing on iNat-120K) with 30% client participation is realistic for FL.

**Methods And Evaluation Criteria:**

The proposed method efficiently estimates within-class covariance without explicitly communicating them from clients, which is technically sound.

The evaluation criteria—accuracy and communication cost—are appropriate for FL, where efficiency and performance under non-iid conditions are critical. Benchmark datasets (CIFAR-100, ImageNet-R, CUB200, Stanford Cars, iNaturalist-Users-120K) span diverse scales and heterogeneity levels, aligning with real-world FL scenarios. The use of Dirichlet distributions (α=0.1) to simulate non-iid data and real-world iNaturalist data enhances relevance.

**Other Comments Or Suggestions:**

- Can you quantify the bias’s practical impact in non-iid scenarios beyond DomainNet?

**Other Strengths And Weaknesses:**

Strength

- Well-structured, with clear explanations of methodology (e.g., Algorithm 1) and results.

Weakness

- Lacks convergence analysis for fine-tuning, weakening claims of improved convergence.

**Questions For Authors:**

- How gamma and lambda are chosen for each benchmark?

**Relation To Broader Scientific Literature:**

FedCOF builds on prior FL work with pre-trained models:
FedNC: FedCOF extends this by estimating covariances from means, improving accuracy without extra communication.
Fed3R: FedCOF reduces communication costs while matching or exceeding performance, addressing a key scalability issue.

**Theoretical Claims:**

Proofs look correct, but the reliance on iid assumptions limits their generalizability, as acknowledged in Appendix E’s bias analysis for non-iid settings.

---

> ### Author Rebuttal · Authors · 2025-03-31
>
> We thank the reviewer for acknowledging the soundness of our work, the correctness of our proofs, and that the evaluation criteria is appropriate for FedL - accuracy, communication cost, FL rounds, and client participation. We also appreciate the recognition of our diverse experimental setup across multiple datasets, which reflect real-world scenarios and enhances the relevance of our work. Finally, we thank the reviewer for highlighting the quality of our estimator and the improved  performance and communication efficiency over existing training-free methods. Below we reply to specific points raised by the reviewer.
>
> >The covariance estimator is unbiased under the iid assumption. Proposition 2 and its proof in Appendix C provide a mathematical foundation, showing that the expectation of the estimator equals the population covariance. However, the evidence weakens when the iid assumption is relaxed (Appendix E), where bias is acknowledged but not quantified beyond theoretical derivation, leaving practical implications partially unsupported. Proofs look correct, but the reliance on iid assumptions limits their generalizability, as acknowledged in Appendix E’s bias analysis for non-iid settings. Can you quantify the bias’s practical impact in non-iid scenarios beyond DomainNet?
>
> In our paper we theoretically analyze the bias of our covariance estimator with non-iid client features (see Appendix E) and perform empirical evaluation on the DomainNet feature shift setting (see Table 5 in Appendix F) to study the impact of the bias.
>
> Based on the Reviewer's suggestion to evaluate the methods in a feature shift setting beyond DomainNet, we perform experiments on the Office-Caltech10 dataset using MobileNetv2 as done in [X]. This dataset contains real-world images obtained from different cameras or environments and has 4 domains (clients) covering 10 classes. Our results are given in the following table:
> |Method|Office-Caltech10|
> |------|------|
> |FedNCM|94.3|
> |Fed3R|94.7|
> |FedCOF|95.3|
>
> In this feature-shift benchmark, we see that all the training-free methods perform similarly. Despite the theoretical bias in our estimator, FedCOF performs similar to Fed3R. We argue that this outcome is due to the generalization performance of pre-trained models. While non-iid feature shift settings have been studied in some papers not using pre-trained models, using this setting with pre-trained models works a bit differently. When using a pre-trained model, the generalization capabilities of the pre-trained model can help in moving the distribution of class features across clients towards an iid feature distribution even if the class distribution across clients is non-iid at the image level.
>
> We observe a dip in performance for FedCOF compared to Fed3R in our experiments on DomainNet (see Table 5 in Appendix F) indicating the effect of the bias in our estimator on that particular benchmark. We believe that more comprehensive analysis of feature-shift settings when using pre-trained models requires more extensive benchmarks and could be an interesting direction to explore in future work. We will add the experiments on Office-caltech10 and this discussion in the revised version.
>
> [X] Li et al., Fedbn: Federated learning on non-iid features via local batch normalization. In ICLR, 2021.
>
> >Justification of using only within class covariance terms are supported only by numerical results (ablative study in Table 2), and this methods underperforms Fed3R in the presence of feature shift. More extensive analysis and justification with theoretical evidence is required for the proposed method.
>
> We discuss this in response to reviewer FJXY. We kindly ask the reviewer to refer to that discussion. We will incorporate this discussion and motivation of excluding between-class covariances in the paper.
>
> ---
>
> >Lacks convergence analysis for fine-tuning, weakening claims of improved convergence.
>
> Our claims of improved convergence using the proposed FedCOF initialization is based on our empirical results (see Fig. 3) across multiple datasets. We propose how to better initialize the classifier before performing federated optimization methods like FedAvg and FedAdam which have already established the theoretical guarantees of convergence in their respective works. We discuss this in detail in the Appendix M.
>
> ---
>
> >How gamma and lambda are chosen for each benchmark?
>
> Fed3R proposed to use $\lambda$ for numerical stability and we use the same value as Fed3R ($\lambda=0.01$) in all our experiments.
>
> Regarding selection of shrinkage hyper-parameter $\gamma$, we do not optimize it for each benchmark and use $\gamma=1$ for all datasets when using SqueezeNet and ViT-B/16 networks. When using MobileNetv2, we use $\gamma=0.1$ due to its very high feature dimensionality (d=1280). We selected hyperparameters on one dataset (ImageNet-R) and used that value for all others, which works well. We discuss the impact of $\gamma$ in Appendix L (see Table 9).

---

### Official Review · Reviewer_6xS7 · 2025-03-18

**Overall Recommendation:** 3

**Summary:**

The paper presents FedCOF – a training-free method that leverages the first-order statistics (class means and variance matrices) to update the global classifier on the backbone of a pre-trained model.

**Claims And Evidence:**

The claim  “the samples belonging to the same class across different clients are sampled from the same distribution” is not trivial and the authors did have mentioned this in appendix F and in Limitations.

**Essential References Not Discussed:**

The work of Luo et. al. [1] was mentioned briefly in the Introduction as a previous work that used the same techniques: class means and covariances from all clients for classifier calibration. However, the author did not include this work (namely, CCVR) as a baseline. Which, in turn, weakens the novelty of the paper.

**Experimental Designs Or Analyses:**

The baseline CCVR that follows the same methodology, is not included in the experiments. I believe this is a significant lack.

**Methods And Evaluation Criteria:**

Yes

**Other Comments Or Suggestions:**

I am not overconfident with my assessment of this work. Based on other reviews and the address of the authors, I will consider changing my score.

**Other Strengths And Weaknesses:**

**Strengths**:

1. The paper is clearly written and easy to follow. The demonstration figure, however, is not. I would recommend replacing this figure with a more comprehensive one in case this work is accepted for publication.

2. The authors notice and exploit the fact that “the samples belonging to the same class across different clients are sampled from the same distribution”. However, this is an overstatement as there are many works dedicated to distribution shifts in Federated Learning where the distribution of same-class samples can differ from client to client (As also shown by the authors in Appendix F and discussed in Limitations). Nonetheless, within the scope of the research, this is but a minor drawback, so I will let it slide.

3. I have an impression that mathematical derivations are basic point-estimations thus I did not pay much attention to the details. However, the overall good experimental results seem to validate equations and formulas.


**Weaknesses**:

1.	The work of Luo et. al. [1] was mentioned briefly in the Introduction as a previous work that used the same techniques: class means and covariances from all clients for classifier calibration. However, the author did not include this work (namely, CCVR) as a baseline. Which, in turn, weakens the novelty of the paper.

**Questions For Authors:**

None

**Relation To Broader Scientific Literature:**

In my opinion, this work does not provide any breakthrough or significant technical contribution to the broader literature. Just a decent work that have positive improvement over some existing works.

**Theoretical Claims:**

I have not checked closely any proof. I did read the sketch proof in the main script.

---

> ### Author Rebuttal · Authors · 2025-03-31
>
> We thank the reviewer for appreciating the clarity and readability of our paper, as well as the good experimental results that validate our mathematical derivations. Below we reply to each of the points raised by the reviewer.
>
> >The work of Luo et. al. [1] was mentioned briefly in the Introduction as a previous work that used the same techniques: class means and covariances from all clients for classifier calibration. However, the author did not include this work (namely, CCVR) as a baseline. Which, in turn, weakens the novelty of the paper.
>
> We thank the reviewer for the suggestion to include CCVR in our comparison. Since CCVR was originally proposed for calibrating classifiers after training, we adapt it to our setting and use it as an initialization method. CCVR trains a linear classifier on features sampled from aggregated class distributions from clients. We show that the proposed FedCOF outperforms CCVR in most settings despite having significantly lower communication cost (in MB):
>
> |Dataset|Method|SqueezeNet| (d=512)||MobileNetv2| (d=1280)||ViT-B/16 |(d=768) |
> |--|--|-------------------------|---------|--|----------------------------|---------|--|-------------------------|---------|
> |||Acc (↑)|Comm. (↓)||Acc (↑)|Comm. (↓)||Acc (↑)|Comm. (↓)
> |**CIFAR100**|CCVR|**57.5**±0.2|3015.3||59.6±0.2|18823.5||72.3±0.2|6780.0|
> | |FedCOF (Ours)|56.1±0.2|**5.9**||**63.5**±0.1|**14.8**||**73.2**±0.1|**8.9**|
> ||
> |**IN-R**|CCVR|36.4±0.2|3645.7||41.9±0.2|22758.8||49.3±0.2|8197.4|
> | |FedCOF (Ours)|**37.8**±0.4|**7.1**||**47.4**±0.1|**17.8**||**51.8**±0.3|**10.7**|
> ||
> |**CUB200**|CCVR|51.2±0.1|2472.1||61.6±0.2|15432.7||78.7±0.4|5558.6|
> | |FedCOF (Ours)|**53.7**±0.3|**4.8**||**62.5**±0.4|**12.0**||**79.4**±0.2|**7.2**|
> ||
> |**Cars**|CCVR|40.9±0.4|2767.3||36.0±0.4|17275.7||49.4±0.4|6222.5|
> | |FedCOF (Ours)|**44.0**±0.3|**5.4**||**47.3**±0.5|**13.5**||**52.5**±0.3|**8.1**|
>
> Note that CCVR does not require federated training since it trains a linear global classifier unlike closed form, training-free approaches.
>
> We also show that FedCOF initialization is better for further finetuning using a pre-trained SqueezeNet that we finetune with FedAdam for 100 rounds after different initialization methods:
> |Method|Training|ImageNet-R|CUB200|Cars|
> |------|--------|----------|------|----|
> |FedNCM+FedAdam|✔|44.7±0.1|50.2±0.2|48.7±0.2|
> |CCVR+FedAdam|✔|44.6+-0.3|51.5+-0.2|47.9+-0.1|
> |Fed3R+FedAdam|✔|45.9±0.3|51.2±0.3|47.4±0.4|
> |FedCOF+FedAdam|✔|**46.0**±0.4|**55.7**±0.4|**49.6**±0.6|
>
> We will add the CCVR baseline in the main experiment tables of our paper in the revised version.
>
> ---
>
> >The paper is clearly written and easy to follow. The demonstration figure, however, is not. I would recommend replacing this figure with a more comprehensive one in case this work is accepted for publication.
>
> We thank the reviewer for the suggestion and will improve the figure with a more comprehensive one in the revised version of the paper.

---

### Official Review · Reviewer_WgDY · 2025-03-20

**Overall Recommendation:** 2

**Summary:**

This paper studied the problem of using pre-trained models to speed up federated learning algorithms by using first-order statistics to estimate second-order statistics to achieve good learning performance without training. The authors proposed a new method to only use first-order statistics in the form of class means communicated by clients to the server, which enjoys low communication costs. The authors showed that these estimated class covariances can be used to initialize a linear classifiers, thus exploiting the covariances without sharing them. The authors performed experiments to illustrate the effectiveness of the proposed method.

**Claims And Evidence:**

The claims made in the submission are justified theoretically and by numerical experiments.

**Essential References Not Discussed:**

None noted.

**Experimental Designs Or Analyses:**

The experimental design and analyses are sound.

**Methods And Evaluation Criteria:**

This paper tested their method on the CIFAR-100 and ImageNet-R, CUB200, StanfordCars, and iNaturalist datasets with SqueezeNet, MobileNetV2, and ViT-B/16 models. They also compared their methods with several training-free and training-based FL methods. The methods and evaluation criteria are sound.

**Other Comments Or Suggestions:**

None.

**Other Strengths And Weaknesses:**

Strengths:

1. Learning-free federated learning with pre-trained models is an interesting and timely topic.

2. This paper conducts comprehensive experiments to verify the performance of the proposed method.

Weaknesses:

1. Most of the theoretical analyses in this paper are rather standard and straightforward. The theoretical contributions of this paper are marginal.

2. The proposed covariance estimator for learning-free FL relies on i.i.d.. assumption class data assumption across clients, which rarely holds true in practice. Although the authors provided theoretical bias analysis and conducted empirical studies, it remains unclear theoretically how non-i.i.d. data could affect the proposed FedCOF method.

3. The comparisons between the proposed FedCOF and FedAvg may not be fair, since one is based on pre-trained models and the other is based on training from scratch.

4. While learning-free FL methods are interesting particularly with pre-trained models, it appears that their use cases are rather limited (e.g., linear classification problems). Could the authors illustrate more relevant use cases for the proposed learning-free FL method?

**Questions For Authors:**

See comments in the weakness section above.

**Relation To Broader Scientific Literature:**

This paper is an interesting contribution to the area of learning-free federated learning methods.

**Theoretical Claims:**

This paper made theoretical claims on the statistical properties of the proposed covariance estimator and the ridge regression solutions based on the estimated class means and covariances. I have checked and verified the correctness of the theoretical results. However, most of the theoretical proofs in this paper are rather standard and straightforward and lack theoretical depth.

---

> ### Author Rebuttal · Authors · 2025-04-01
>
> We thank the reviewer for acknowledging that our claims are justified both theoretically and through numerical experiments, for recognizing the correctness of our theoretical results, the soundness of our experimental design and analyses, the comprehensiveness of our evaluation, and for appreciating that we address the interesting and timely topic of training-free methods, providing an interesting contribution. Below, we respond to each point raised by the reviewer.
>
> >Most of the theoretical analyses in this paper are rather standard and straightforward. The theoretical contributions of this paper are marginal.
>
> While our analysis builds on established mathematical tools, it provides non-trivial theoretical insights crucial for the effectiveness of our method and relevant to general-purpose Federated Learning.
>
> Specifically, we rigorously derive an unbiased estimator of class covariances using only first-order statistics (Prop. 2), enabling the estimation of second order statistics without sharing them. This approach avoids higher communication costs and mitigates privacy concerns. We believe this novel theoretical result is impactful for training-free method initialization, as demonstrated by our results. Moreover, it can be used in broader federated learning contexts where second-order statistics are needed but costly to transfer since  communication costs can be bottleneck for real-world FL systems.
>
> Additionally, in Prop. 3 we perform novel derivation of the Ridge Regression solution that, when combined with our estimator, avoids the need to share large matrices, thereby significantly reducing communication costs. We emphasize that this derivation is not straightforward, as also acknowledged by Reviewer diuQ, and, to the best of our knowledge, we are the first to provide it.
>
> > Although the authors provided theoretical bias analysis and conducted empirical studies, it remains unclear theoretically how non-i.i.d. data could affect the proposed FedCOF.
>
> We acknowledge that our covariance estimator for learning-free FL relies on standard i.i.d assumption across clients -- a common assumption in most of FL literature -- and we explicitly state this limitation in our paper. In Appendix E, we provide a detailed derivation of the bias introduced when this assumption is violated:
> $\text{Bias}(\hat{\Sigma}) = \mathbb{E}[\hat{\Sigma}] -  \Sigma = \frac{1}{K-1} \sum_{k=1}^K (\Sigma_k - \Sigma) + \frac{1}{K-1}\left(\sum_{k=1}^K n_k (\mu_k -\mu )(\mu_k -\mu)^\top \right).$
>
> This derivation quantifies the bias that arises when the distribution of a class within a client differs from the global distribution of the same class. In Appendix F we empirically demonstrate the impact of this bias in feature-shift setting using DomainNet.
>
> While we have focused our theoretical analysis on quantifying the bias, our extensive empirical evaluation on large-scale real-world non i.i.d benchmark -- such as iNaturalist-120k, having 1203 classes across 9275 clients, 120k training images of natural species collected around the world -- further demonstrates that our method is robust (see Table 3 and Fig. 4). We will consider exploring further theoretical implications of non-i.i.d data in future work.
>
> >The comparisons between the proposed FedCOF and FedAvg may not be fair, since one is based on pre-trained models and the other is based on training from scratch.
>
> We would like to clarify that *all FedAvg and FedAdam experiments in the main paper (See Figs. 3-4 and Table 4) use pretrained backbones and are not trained from scratch, thus ensuring fair comparison with FedCOF.* We have more experiments in the Appendix (see Table 8) in which we compare FL methods with and without pre-trained backbone initialization.
>
> >While learning-free FL methods are interesting particularly with pre-trained models, it appears that their use cases are rather limited (e.g., linear classification problems). Could the authors illustrate more relevant use cases for the proposed learning-free FL method?
>
> We would like to highlight that learning-free FL methods use cases are not limited to linear classification problems. Their purpose is to exploit the feature distribution given by a pre-trained model to initialize a classifier. Training-free classifier initialization can be followed by full federated finetuning to solve non-linear classification tasks (see Fig. 3). We demonstrate that this approach (FedCOF+FedAdam, Fig. 3) can improve performance at a much lower costs compared to federated full finetuning, with pre-trained backbone, and a classifier randomly initialized (FedAvg, FedAdam, Fig. 3).
>
> Finally, training-free methods could be used in other tasks like object detection or semantic segmentation. As an example, object detection networks like Faster-RCNN use a classification head and prototype-based closed form approaches could be adapted to those network heads.

---

### Official Review · Reviewer_diuQ · 2025-03-24

**Overall Recommendation:** 3

**Summary:**

This paper introduces FedCOF, a training-free FL framework that seeks to compute (in FL fashion) a closed form ridge regression solution using features extracted from a pre-trained model. The naive solution to this formulation, which was done in Fed3R, requires sharing second-order statistics which, in the context of FL, consumes significant bandwidth. This paper presents an analysis showing that the closed form solution can be estimated using only feature mean from each class. Empirical results show positive improvement (in terms of both performance and communication bandwidth) over other training-free and training-based FL solutions.

**Claims And Evidence:**

The motivation for this paper is largely based on a previous line of research on FL with a pre-trained model (FedNCM). At the time, I believe training-free FL is shown to have better performances than FL-training of a classification head/ FL-training from scratch. Federated full-finetuning was not considered due to the expensive cost. However, since then there have been some advances in federated finetuning, such as federated LoRA and federated fine-tuning. I'm not sure if the argument of FedNCM still hold without being re-positioned against newer techniques.

A shortcoming of this method (and its predecessor Fed3R) is that it is confined to a ridge regression head, which is where it derives the closed form solution from. However, ridge regression may not be the best choice for complex task. For example, looking at Fig. 3 of this paper, we can see that the accuracy continues to increase after the training-free stage of FedCOF/Fed3R/FedNCM, which should not be the case if ridge regression is reasonably good & we can compute its closed form solution.

The main claims of this paper are correct, which is expected since Propositions 1 and 2 are well known statistical results. Prop 3 is a clever rewrite of the ridge regression solution, which neatly avoids the need to share large matrices.

**Essential References Not Discussed:**

- Probabilistic Federated Prompt-Tuning with Non-IID and Imbalanced Data (Weng et al., 2024)
- Heterogeneous LoRA for Federated Fine-tuning of On-device Foundation Model (Cho et al., 2023)
- FedPrompt: Communication-Efficient and Privacy Preserving Prompt Tuning in Federated Learning (Zhao et al., 2022)

**Experimental Designs Or Analyses:**

Experimental design generally makes sense. I have pointed out some baselines that need to be compared with.

**Methods And Evaluation Criteria:**

I strongly believe this paper should compare against parameter efficient federated fine-tuning approaches. I believe it has been pointed out previously that tuning only a classification head is far inferior to these techniques. Some baselines to consider:
- Probabilistic Federated Prompt-Tuning with Non-IID and Imbalanced Data (Weng et al., 2024)
- Heterogeneous LoRA for Federated Fine-tuning of On-device Foundation Model (Cho et al., 2023)
- FedPrompt: Communication-Efficient and Privacy Preserving Prompt Tuning in Federated Learning (Zhao et al., 2022)

In terms of performance report, I notice that the performance of FedAvg after 200 rounds of comm is generally not converged. It would be interesting to see if FedAvg would eventually outperform both the training free performance and the training free + finetuning performance. Regardless, I still see the merit of this method as an inexpensive way to initialize the classifier, which could save 100 or more communication rounds in practice.

**Other Comments Or Suggestions:**

I have no further comments

**Other Strengths And Weaknesses:**

I have no further comments

**Questions For Authors:**

I have no further questions

**Relation To Broader Scientific Literature:**

The paper is missing a body of literature on federated parameter efficient fine-tuning, which should be a more appropriate competitor than classification head fine-tuning.

**Theoretical Claims:**

I checked all three propositions. They are sound.

---

> ### Author Rebuttal · Authors · 2025-03-31
>
> We thank the reviewer for acknowledging the soundness of our propositions, the positive improvements in performance and communication cost over other training-free and training-based solutions, and the quality of our experimental design. We appreciate that the reviewer acknowledged that Proposition 3 is a clever rewrite of the ridge regression solution, which neatly avoids the need to share large matrices. Below we address the specific points raised by the reviewer.
>
> > Federated full-finetuning was not considered due to the expensive cost.
>
> We would like to clarify that our paper considers federated full-finetuning for comparison with FedCOF (See Fig. 3 and Table 4). We consider FedAvg and FedAdam as the federated full-finetuning baselines starting from a pretrained model initialization. All methods denoted as FedAvg and FedAdam in the main paper perform federated training from pretrained model initialization in which **only** the classifier is randomly initialized without using training-free initialization methods. It is important to note that these models are not trained from scratch. We have some experiments in Appendix K (see Table 8) in which we compare federated training methods with and without pretrained backbone initialization.
>
> > I strongly believe this paper should compare against parameter efficient federated fine-tuning approaches. The paper is missing a body of literature on federated parameter efficient fine-tuning, which should be a more appropriate competitor than classification head fine-tuning.
>
> We thank the reviewer for suggesting recent works on parameter-efficient federated fine-tuning. We will discuss these papers in the revised version. For comparison with these methods, we consider the recent work Probabilistic Federated Prompt-Tuning (PFPT) from NeurIPS 2024, as suggested by the reviewer. We compare PFPT, FedAvg-PT, and FedProx-PT (prompt-tuning variants of FedAvg and FedProx) with training-free initialization approaches, and also perform PFPT with training-free  classifier initialization.
>
> We use a pre-trained ViT-B32 on CIFAR-100 and TinyImageNet with a Dirichlet distribution ($\alpha=0.1$) following PFPT. We use the same training hyperparameters as PFPT. The results are summarized in the following table:
> ||Training-Free|CIFAR-100|CIFAR-100||Tiny-ImageNet|Tiny-ImageNet|
> |:-:|:-:|:-:|:-:|:-:|:-:|:-:|
> |||Accuracy (↑)|Comm. (↓, MB)||Accuracy (↑)|Comm. (↓, MB)|
> |FedAvg-PT|❌|74.50|717.2||78.58|1410.4|
> |FedProx-PT|❌|73.60|717.2||79.19|1410.4|
> |PFPT|❌|75.08|713.4||82.31|1415.1|
> ||||
> |FedNCM|✅|67.70|8.9||76.93|17.8|
> |Fed3R|✅|75.24|244.8||81.51|246.6|
> |**FedCOF (ours)**|✅|75.23|8.9||81.62|17.8|
> ||||
> |FedNCM+PFPT|❌|75.70|722.3||86.18|1432.9|
> |Fed3R+PFPT|❌|76.58|961.3||86.41|1661.7|
> |**FedCOF (ours)+PFPT**|❌|76.67|722.3||86.26|1432.9|
>
> Fed3R and the proposed FedCOF are competitive with the prompt-based training methods, without any training. Training-free methods require lower communication budget with respect to prompt-tuning methods. Finally, these results show that further finetuning of prompts after training-free initialization with Fed3R and FedCOF achieves state-of-the-art results on these benchmarks.
>
> >Ridge regression may not be the best choice for complex task. For example, looking at Fig. 3 of this paper, we can see that the accuracy continues to increase after the training-free stage of FedCOF/Fed3R/FedNCM, which should not be the case if ridge regression is reasonably good & we can compute its closed form solution.
>
> In Fig 3. all methods perform federated full-finetuning starting from a pre-trained backbone with the classifier initialized randomly (FedAvg, FedAdam) or with a training-free method. This improves the quality of feature representations and thus the classification performance improves. Instead if we consider Fig. 4, where we do training-based federated linear-probing (keeping the backbone fixed), we see that the performance improves only marginally after the Fed3R and FedCOF suggesting that they are pretty good classifier. While linear probing marginally improves performance, it requires training (linear-probing) for several rounds and incurs larger communication and computational costs.
>
> > It would be interesting to see if FedAvg would eventually outperform both the training free performance and the training free + finetuning performance. Regardless, I still see the merit of this method as an inexpensive way to initialize the classifier, which could save 100 or more communication rounds in practice.
>
> We compare FedAvg trained for 400 rounds with FedCOF+FedAdam trained for 100 rounds and observe that FedAvg which converges after 400 training rounds still does not outperform FedCOF+FedAdam:
> |Method|ImageNet-R|CUB200|Cars|
> |--|--|--|--|
> |FedAvg (100 rounds)|30.0|23.5|24.8|
> |FedAvg (400 rounds)|41.3|49.3|49.4|
> |FedCOF+FedAdam (100 rounds)|46.0|55.4|49.3|
>
> In the revised version of the manuscript, we will include the plot of FedAvg for 400 rounds.

---

### Decision · Program_Chairs · 2025-05-01

**Decision:**

Reject

**Comment:**

Overall, taking into account both the ratings and the written comments from all reviewers, I believe this paper is in the borderline.

It has a nice idea on training-free initialization for federated learning with pre-trained model. I think the reviewing panel likes this idea. But, unfortunately, it was not adequately positioned against an existing literature on federated fine-tuning which also uses pre-trained models. There are also ambiguities in the application of its core principle to cross-device setting, as well as its empirical comparison with other approaches on communication cost. These are summarized below.

--

First, both reviewers (WgDY, diuQ) have pointed out the lack of comparisons with several recent methods in federated fine-tuning. I see that the authors have run extensive comparison with one prompt-based method showing some interesting insight on the low communication cost of the proposed training-free initialization method. Reviewer diuQ therefore remains supportive of this paper but the concern raised by WgDY was however not addressed. There is currently no comparison with federated LoRA -- see [*]

[*] https://arxiv.org/abs/2409.05976 (NeurIPS 2024)

Second, for the LoRA-based method, the communication cost per round is O(rd) where r is the rank of the LoRA adapter where r is usually small (i.e., 4-16) so suppose there are T rounds of communication with m << K clients per round, the cost is O(Tmrd).

In comparison, FedCOF's communication cost of its multiple-round version (see lines 311-322 for cross-device settings) will be O(TmCd) where T is the no. of communication iterations. It is therefore unclear if FedCOF is more advantageous than federated LoRA in terms of communication cost.

Even in comparison to the prompt-based PFPT, the advantage is still somewhat ambiguous. For example, the communication per round of PFPT is m * p * O(d) where p is the no. of prompt selected per client and so its overall cost is O(Tmpd) so given the huge communication gap between FedCOF and PFPT, it's likely that PFPT uses a lot of prompts per client which might be more than needed. The comparison against PFPT could have been more compelling if the results are ablated across different choices of p for PFPT.

Third, it seems the communication cost of FedCOF in the experiment against PFPT is for the single-round setting where all clients tune in whereas in PFPT, it seems only 10/100 clients tune in per iteration. If this is true, the comparison with PFPT is not quite apple-to-apple (i.e., different data partition/simulation settings).

In addition, even if all clients participate in a single round, the cost will be O(dCK) where d = 768, C = 100, K = 100 for the CIFAR100 dataset. Assuming each number is 32-bit as indicated in the paper, shouldn't the communication be more than 8.9MB? Also note that for federated learning in general, we might be able to reduce the no. iterations via increasing the number of local updates which in turn reduces communication cost.

Fourth, in cross-device setting, FedCOF (as described in lines 311-322) cannot expect all clients to participate in a single round so unless there is a tracking mechanism, FedCOF will likely accumulate duplicate info from the same clients across iterations. Without a proper tracking mechanism, even when FedCOF has acquired information from all clients, how do we aggregate them to generate unbiased estimators? Otherwise, tracking which update belong to which client at which iteration will likely violate each client's differential privacy.

--

In summary, while the method is indeed interesting, it has (1) not been adequately positioned against existing literature on federated fine-tuning, lacking comparison with relevant federated LoRA baselines; and (2) multiple ambiguities in both experiments (communication cost comparison) and algorithmic principles (e.g., how to guarantee unbiasedness in cross-device settings).

--

Given the above, I unfortunately cannot bump this paper over the borderline. I hope the authors will revise the paper accordingly to strengthen it for future submission.